# Design and evaluation of a pneumatic actuation unit for a wasp-inspired self-propelled needle

**Jette Bloemberg**[1]*, **Bruce Hoppener**[1], **Bram Coolen**[2], **Aimée Sakes**[1], **Paul Breedveld**[1]

**1** Faculty of Mechanical Engineering, Department of BioMechanical Engineering, Bio-Inspired Technology (BITE) Group, Delft University of Technology, Delft, The Netherlands, **2** Department of Biomedical Engineering & Physics, Amsterdam UMC Location University of Amsterdam, Amsterdam, The Netherlands

* J.Bloemberg@tudelft.nl

**Data Availability Statement:** All relevant data are within the manuscript and its Supporting Information files.

**Funding:** This work was supported by the Netherlands Organisation for Scientific Research

## Abstract

Transperineal laser ablation is a minimally invasive thermo-ablative treatment for prostate cancer that requires the insertion of a needle for accurate optical fiber positioning. Needle insertion in soft tissues may cause tissue motion and deformation, resulting in tissue damage and needle positioning errors. In this study, we present a wasp-inspired self-propelled needle that uses pneumatic actuation to move forward with zero external push force, thus avoiding large tissue motion and deformation. The needle consists of six parallel 0.25-mm diameter Nitinol rods driven by a pneumatic actuation system. The pneumatic actuation system consists of Magnetic Resonance (MR) safe 3D-printed parts and off-the-shelf plastic screws. A self-propelled motion is achieved by advancing the needle segments one by one, followed by retracting them simultaneously. The advancing needle segment has to overcome a cutting and friction force, while the stationary needle segments experience a friction force in the opposite direction. The needle self-propels through the tissue when the friction force of the five stationary needle segments overcomes the sum of the friction and cutting forces of the advancing needle segment. We evaluated the prototype's performance in 10-wt% gelatin phantoms and *ex vivo* porcine liver tissue inside a preclinical Magnetic Resonance Imaging (MRI) scanner in terms of the slip ratio of the needle with respect to the phantom or liver tissue. Our results demonstrated that the needle was able to self-propel through the phantom and liver tissue with slip ratios of 0.912–0.955 and 0.88, respectively. The prototype is a promising step toward the development of self-propelled needles for MRI-guided transperineal laser ablation as a method to treat prostate cancer.

## 1. Introduction

### 1.1 Transperineal laser ablation

For patients diagnosed with prostate cancer, there are multiple treatment options, including radical surgery, radiotherapy, and systemic treatment, such as chemotherapy, hormonal therapy, or immunotherapy [1]. Treatments targeting the entire gland can result in unwanted side

(Nederlandse Organisatie voor Wetenschappelijk Onderzoek, NWO), domain Applied and Engineering Sciences (TTW), and which is partly funded by the Ministry of Economic Affairs. Grant number 80450, Perspectief programme, Photonics Translational Research – Medical Photonics (MEDPHOT), awarded to PB. URL: https://www.nwo.nl/. The funders had no role in study design, data collection and analysis, decision to publish, or preparation of the manuscript.

**Competing interests:** The authors have declared that no competing interests exist.

effects due to damage to the surrounding healthy tissue [1–3]. To address this issue, one potential approach is to apply focal therapy [4–6], a method that targets only the cancerous cells (i.e., the lesion) while safeguarding the adjacent healthy tissue. Trans Perineal Laser Ablation (TPLA) is one such focal therapy with fewer side effects than whole gland options [7]. TPLA is a novel minimally invasive thermo-ablative technique that induces cell death using optical fibers that are inserted into the prostate through transperineally positioned needles [8]. Clinical trials have demonstrated promising results in inducing tissue ablation with minimal treatment-related side effects [7–9].

For TPLA, Magnetic Resonance Imaging (MRI) guidance offers several advantages over guidance from other imaging modalities such as ultrasound, including superior soft tissue contrast for visualizing tumors and real-time temperature monitoring [9–12]. These advantages motivate the development of new Magnetic Resonance (MR) safe or MR conditional needles. According to the American Society for Testing and Materials (ASTM) standard F2503-13 [13], MR safe devices pose no known hazards resulting from exposure to an MR environment, whereas MR conditional devices pose no known hazards resulting from exposure to a specified MR environment with specified conditions of use.

## 1.2 MRI compatible needles

Due to its constricted workplace and strong magnetic field with oscillating gradients, an MRI scanner poses unique challenges to medical instrument engineers designing MR safe or MR conditional needles and physicians performing the prostate interventions when the prostate is near the MRI scanner's isocenter. A number of researchers, such as Cepek *et al.* [14,15] and Bloemberg *et al.* [16], have developed manually actuated systems, which allow the needle(s) to be positioned while the patient is inside the MRI bore. However, in these systems, the physician is required to work within the confined space of the MRI bore. Another solution is keeping the physician at a distance using an automated needle system that is actuated by a technology that does not use metallic, magnetic, or conductive materials [13].

A number of MR safe/conditional actuation methods have been proposed and demonstrated, such as piezomotors [17], Bowden cables [18,19], hydraulics [20], and pneumatics [21–25]. In a hospital setup, pneumatics is advantageous as pressurized air is commonly available in an MRI room and can be controlled with a standard pneumatic valve manifold. An important drawback of pneumatics, however, is the compressibility of air, which means that the only well-defined pneumatic actuator positions are the start and end positions [26]. Nevertheless, this inherent characteristic of pneumatics makes it suitable for step-wise position control [27].

Over the years, various research groups have developed pneumatic actuation systems for needles with the potential for use in MRI-guided interventional procedures. For example, Stoianovici *et al.* [21] developed an actuation system for transperineal prostate biopsy, the MrBot, which is driven by six rotational stepper actuation systems [28]. In addition, Groenhuis *et al.* [27] developed both linear and rotational teeth geometry stepper actuation systems using three-dimensional (3D)-printed parts and seals for a breast biopsy application.

In these pneumatically actuated needles, the actuation systems are designed for conventional needles that are pushed through the tissue, which requires an axial force on the needle. When this axial force exceeds the needle's critical load, the needle will deflect laterally—a mechanical failure mode known as buckling [29]. The lateral deflection poses a risk of damaging tissue in the vicinity of the needle and can result in poor control of the needle's path [30,31]. To reduce tissue damage during needle insertion and allow for accurate needle positioning, wasp-inspired needles have been developed that can be advanced without being

pushed through the tissue [32–35]. In the scientific literature, other sources of bio-inspiration, such as the mosquito [36,37], were used for needles employing different working principles [38]. Furthermore, wasp-inspired propulsion has also been shown to be useful in a drilling device for medical applications [39].

### 1.3 Wasp-inspired needles

Female parasitic wasps use their ovipositor to deposit their eggs into hosts, which may hide in a dense material such as wood [40]. The ovipositor of the parasitic wasp species *Diachasmimorpha longicaudata* (Hymenoptera: Braconidae) is very long (5.7±0.6 mm) [41] and thin (30–50μm) [42] and comprises three slender, parallel segments referred to as valves [43], which reciprocate by advancing and retracting them with respect to each other (Fig 1). The advancing and retracting forces generate a net push force near zero, allowing for self-propulsion within a substrate.

Wasp-inspired self-propelled needles apply this self-propulsion principle to advance through tissue. They are composed of parallel segments that can slide along each other (Fig 2A). Inserting a needle into tissue results in forces acting on the needle by the surrounding tissue. Okamura *et al.* [44] demonstrated that these forces are the sum of the surface stiffness force ($\mathbf{F}_{stiff}$), cutting force ($\mathbf{F}_{cut}$), and friction force ($\mathbf{F}_{fric}$). To insert a needle into tissue, the operator should apply an insertion force ($\mathbf{F}_{in}$) that overcomes the sum of these forces acting on the needle.

The surface stiffness force ($\mathbf{F}_{stiff}$) arises at the needle tip and can be described as the pre-puncture force and is due to the elasticity of the tissue layer and occurs when puncturing the skin or a stiffer tissue layer than the current surrounding tissue, e.g., when puncturing the membrane around an organ [45]. In this pre-puncture phase, the needle is not cutting the tissue, but instead compressing it until it punctures the skin or surface membrane [46]. In the post-puncture phase, after the needle has punctured the tissue layer, the surface stiffness force returns to zero.

Both the cutting and friction forces are post-puncture forces. The cutting force arises at the needle tip when slicing through the tissue due to the plastic deformation of the tissue and the tissue stiffness experienced at the tip of the needle because the needle still encounters stiffness as it cuts through the tissue [47]. When assuming the tissue structure is homogeneous, the

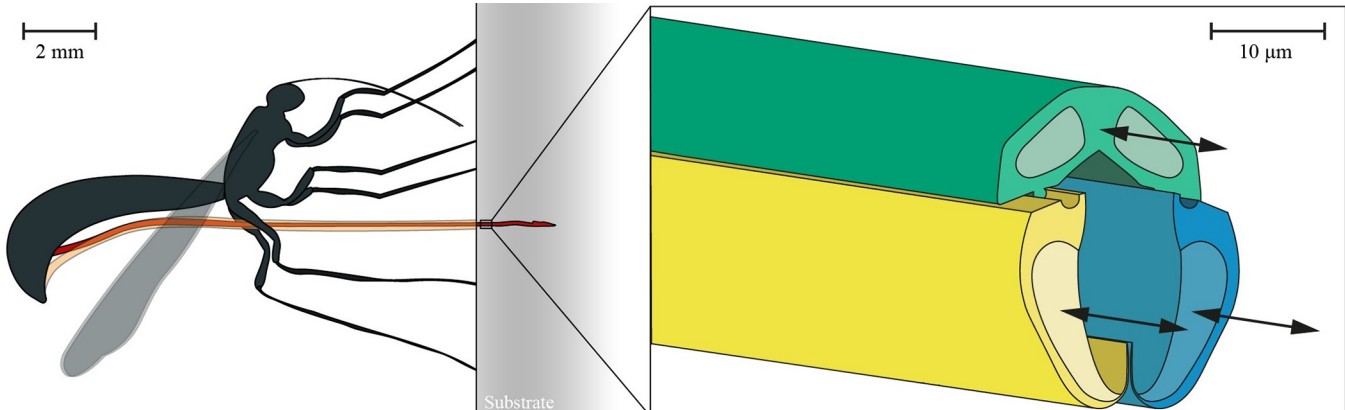

**Fig 1. Visualization of the ovipositor of a female parasitic wasp *Diachasmimorpha longicaudata* (Hymenoptera: Braconidae).** Sheaths (orange) support the ovipositor (red) outside the substrate (gray) that she is probing in. The ovipositor consists of three parallel valves (green, yellow, blue) that can move reciprocally (based on Cerkvenik *et al.* [43]).

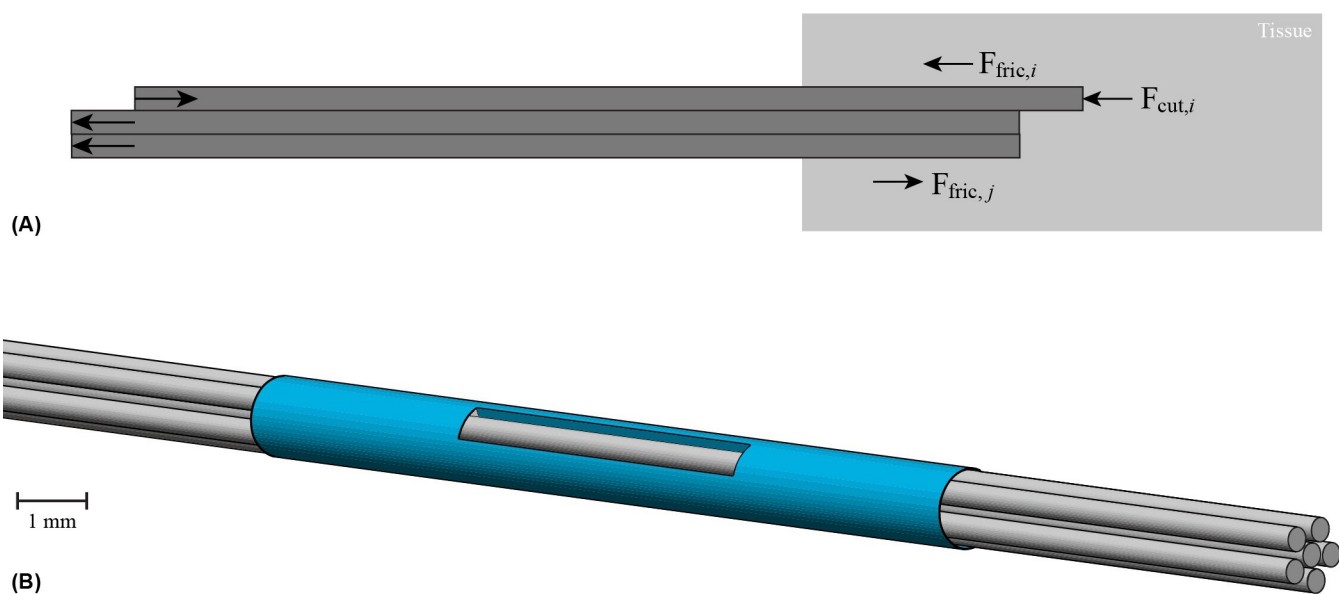

**Fig 2. Visualizations of ovipositor-inspired needles.** (A) Schematic two-dimensional (2D) illustration of ovipositor-inspired needle insertion into tissue with one advancing needle segment and two retracting needle segments in the post-puncture phase after the needle has penetrated the outer tissue layer, therefore, $\mathbf{F}_{\text{stiff}}$ is not present. $\mathbf{F}_{\text{fric},i}$ is the friction force along the advancing needle segment, $\mathbf{F}_{\text{cut},i}$ is the cutting force on the tip of the advancing needle segment, and $\mathbf{F}_{\text{fric},j}$ is the friction force along the retracting needle segments, which works in the opposite direction as the friction force of the advancing needle segments. (B) Schematic three-dimensional (3D) illustration of the ovipositor-inspired needle consisting of six parallel segments (gray) that can slide along each other and are bundled by a shrink tube (blue).

cutting force remains roughly constant throughout the insertion [48]. The friction force occurs along the length of the needle inside the tissue and arises due to Coulomb friction, adhesive friction, and viscous friction [49]. The Coulomb friction force (i.e., static friction force) depends linearly on the normal force acting on the needle body as a reaction to the compression applied by the needle that compresses the tissue out from the needle path, which is affected by the needle diameter and the coefficient of friction between the needle and the tissue [49]. The adhesive friction force is caused by the surface roughness of the needle, which is caused by the tendency of the tissue to stick to the needle surface. The viscous friction force (i.e., dynamic friction force or damping) depends on the damping force during needle translation through the tissue and is proportional to the relative velocity between the needle and surrounding tissue [50].

The self-propelled motion of the needle is accomplished by balancing the cutting and friction forces of the advancing segments with the friction force generated by the remaining stationary or retracting segments [35]. Eq 1 represents the conditions required for the self-propulsion of the needle [51].

$$\sum_{i=1}^{a} \left( \mathbf{F}_{\text{stiff},i} + \mathbf{F}_{\text{fric},i} + \mathbf{F}_{\text{cut},i} \right) \leq \sum_{j=1}^{r} \left( \mathbf{F}_{\text{fric},j} \right) \tag{1}$$

where $a$ refers to the number of advancing needle segments, $r$ refers to the number of retracting or stationary needle segments, $a+r$ is the total number of needle segments, and $\mathbf{F}_{\text{stiff}}$, $\mathbf{F}_{\text{fric}}$, and $\mathbf{F}_{\text{cut}}$ represent the surface stiffness, friction, and cutting forces, respectively. $\mathbf{F}_{\text{stiff}}$ is only present when puncturing tissue layers and is not present whilst self-propelling through homogeneous tissue [51]. The self-propelled motion of the needle requires the friction force generated by the retracting or stationary needle segments to overcome the combined friction and cutting forces of the advancing needle segments. By doing so, the needle is able to gradually

move forward through the tissue as a whole with a self-propelled motion. As the needle propels deeper into the tissue, the surface area of the needle segments in direct contact with the tissue increases, increasing the friction forces on the needle segment linearly with insertion depth while the cutting force remains roughly constant throughout the insertion [48]. Scali *et al.* [51,52] and Bloemberg *et al.* [16] developed various wasp-inspired self-propelled needles with a submillimeter external diameter consisting of three to seven parallel segments that can slide along each other and are bundled by a ring or a shrink tube. The needles self-propel with zero external push force without buckling, by advancing one needle segment at a time over a short distance and slowly retracting the other segments, thereby the friction force of the stationary needle segments overcomes the sum of the friction and cutting forces of the advancing needle segment [16,51].

### 1.4 Goal of this research

Wasp-inspired self-propelled needles could allow for accurate needle positioning for targeted medical MRI-guided procedures to treat prostate cancer, while avoiding unwanted tissue damage. Current prototypes of wasp-inspired self-propelled needles employ either electric motors [35,51,53,54] or manual activation [16] to actuate individual needle segments. Electric motors are not suitable for MRI-guided procedures as they employ metallic, magnetic, and conductive materials, which makes the needles MR unsafe [13]. Manual activation is less suitable for MRI-guided procedures as it requires the physician to work manually within the confined space of the MRI bore. In this research, we propose a pneumatic actuation unit for a wasp-inspired needle that enables physicians to operate the needle under MRI guidance without having to manually actuate the needle within the confined space of the MRI bore.

## 2. Design

### 2.1 Needle

The complete design, called the Pneumatic Ovipositor Needle, consists of a needle, an actuation unit, and a control unit. Following our previous design [16], we decided to focus our research on a self-propelled wasp-inspired needle consisting of *six 200-mm long parallel needle segments* to reach the prostate transperineally (Fig 2B) [55]. The needle self-propels through the substrate by first moving the six needle segments forward one by one (i.e., the advancing phase), followed by moving all six needle segments backward simultaneously (i.e., the retracting phase). In this study, the two-phase motion sequence, which involves advancing the six needle segments one by one followed by retracting all six needle segments simultaneously, will be referred to as a "cycle." The distance that each needle segment travels per cycle is called the "stroke length."

### 2.2 Actuation unit

The actuation unit consists of six pneumatic cylinders, six needle clamps, and a cone. The six pneumatic cylinders are positioned in a circle, so each piston has the cross-sectional shape of $1/6^{th}$ of a circle. Fig 3 shows the working principle of one pneumatic cylinder. The mechanism is simplified and visualized in a two-dimensional (2D) schematic cross-sectional illustration to explain the working principle. The pneumatic cylinder consists of a piston (in green), a piston housing (in gray), a piston stop (in orange), and a piston guide (in red). Each piston can translate in the positive and negative x-direction driven by two pneumatic tubes connected to two separate inlets. By pressurizing Inlet 1, the piston moves in the positive x-direction (Fig 3A)

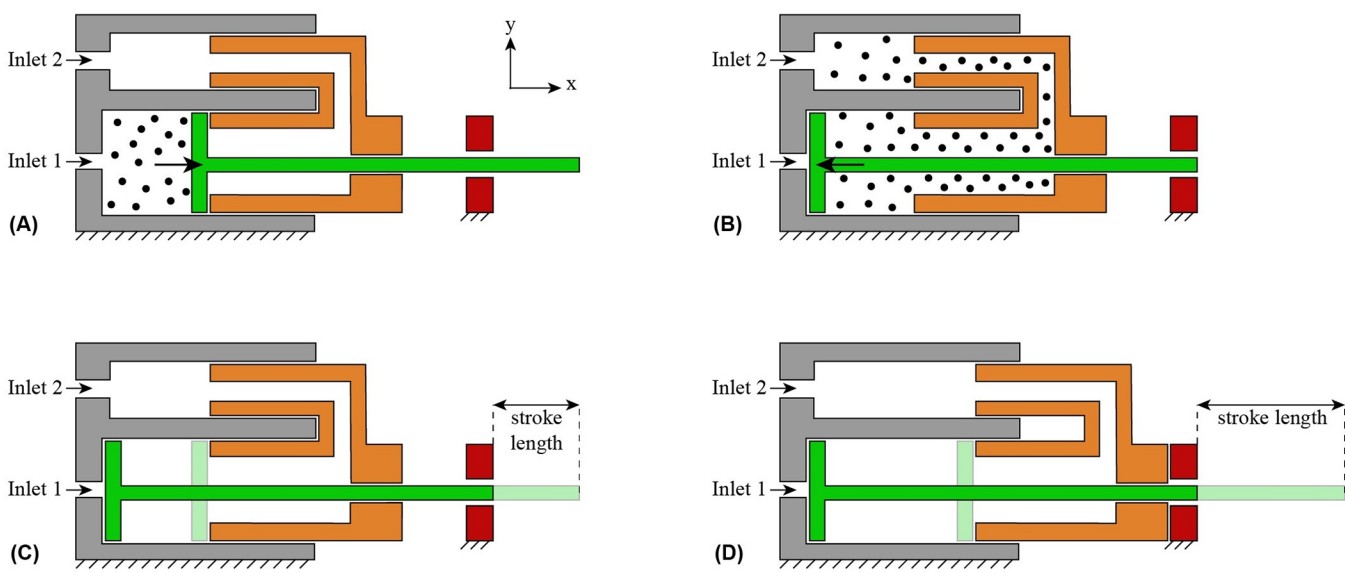

**Fig 3. Schematic illustration of the pneumatic cylinder design composed of the piston housing (gray), piston (green), piston stop (orange), and piston guide (red).** (A) When Inlet 1 is pressurized, the piston translates in the positive x-direction. (B) When Inlet 2 is pressurized, the piston translates in the negative x-direction. The piston stop can translate along the x-axis to allow for an adjustable stroke length between 2 mm (C) and 10 mm (D). The piston stop can be locked in place by tightening screw fasteners (not shown) in the piston guide and piston stop.

and by pressuring Inlet 2, the air will flow through the piston stop, moving the piston in the negative x-direction (Fig 3B).

The piston stop, which fits into the piston housing and is constrained from sliding out of the housing by the piston guide, allows adjustment of the piston stroke length. The piston stop can be moved and locked in place by screw fasteners in the piston guide and piston stop. For a short stroke length, the piston stop is moved and locked at a position close to the piston housing (i.e., in the negative x-direction) (Fig 3C), whereas for a long stroke length, the piston stop is moved and locked at a position far from the piston housing (i.e., in the positive x-direction) (Fig 3D). The stroke length can be varied in the range of 2–10 mm.

To facilitate easy attachment and detachment of the needle segments to the pistons, piston tips (Fig 4A in yellow) were created. Each piston tip has two openings on the distal and proximal sides for securing the piston and needle segment using screw fasteners, respectively.

The needle segments run through the actuation unit at a larger diameter than at the needle tip. A cone (in blue), at the distal side of the actuation unit, smoothly guides the needle segments from the actuation unit to the needle tip through S-shaped channels, which allow the needle segments to slide back and forth freely while avoiding buckling. To prevent blockage of the channels during the stereolithography 3D-printing process, we composed the cone out of two parts (Fig 4B), each containing semi-cylindrical grooves rather than a single part containing cylindrical channels.

### 2.3 Control unit

Pressurized air is required to actively translate the pistons of the actuation unit. The pressure in each cylinder chamber is controlled by a system of electromagnetic valves of type *Festo MHP2-MS1H-5/2-M5* (Festo AG & Co. KG, Esslingen am Neckar, Germany), supplied with a system (gauge) pressure (*P*). The 5/2-way valves have two outputs, when no signal is sent to the valve, the first output, which we connected to Inlet 2, is pressurized, moving the piston in the negative x-direction, thereby retracting the needle segment. When applying an electrical

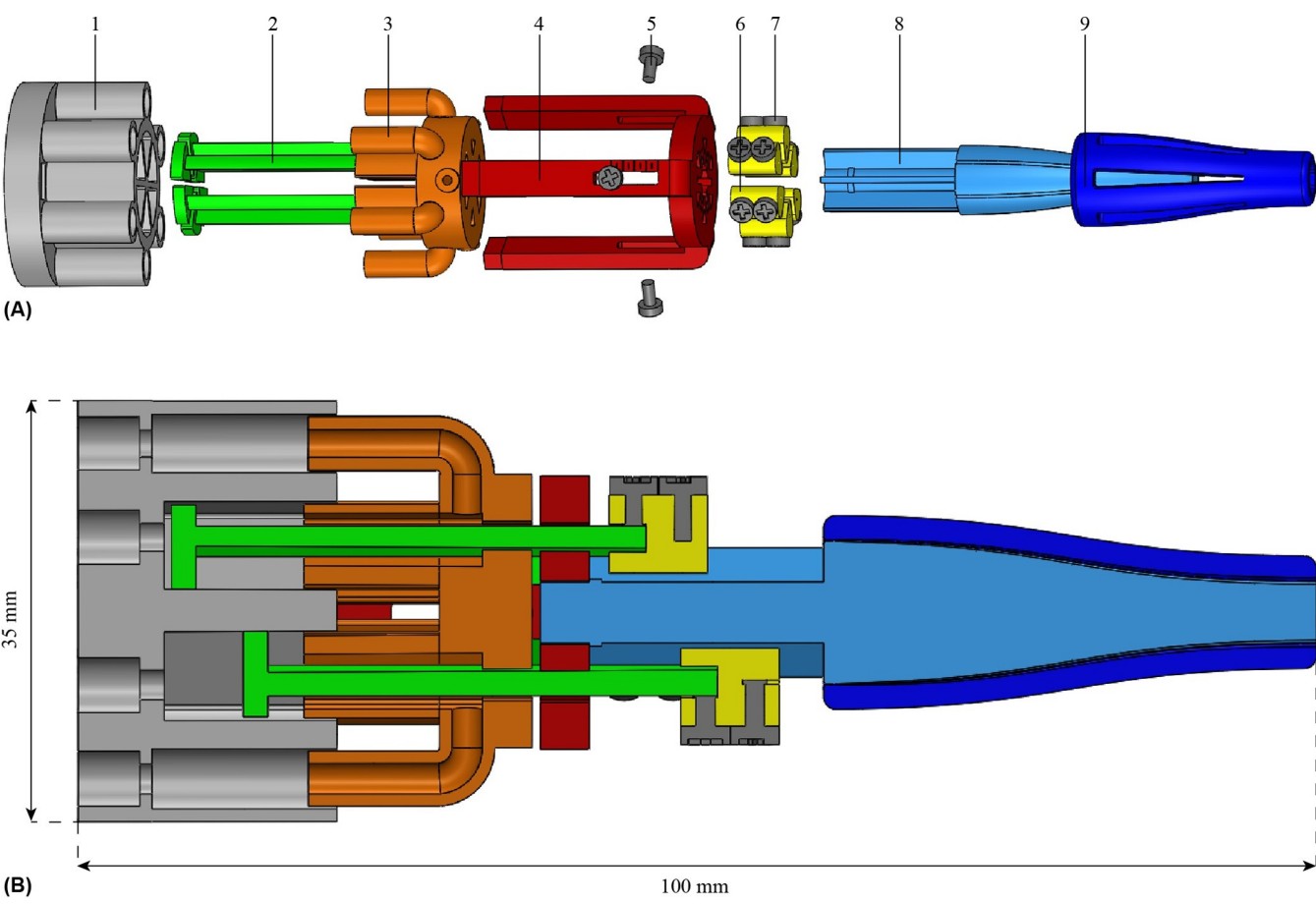

**Fig 4.** Exploded view (A) and cross-section (B) of the entire pneumatic actuation unit, consisting of a piston housing (1), piston (2), piston stop (3), piston guide (4), screws to secure the piston stop (5), piston tip (6), screws to secure the piston tip to the piston and the needle segment (7), inner part of the cone (8), and outer part of the cone (9).

signal, the second output, which we connected to Inlet 1, is pressurized, moving the piston in the positive x-direction, thereby advancing the needle segment. The six valves are controlled by an *Arduino Uno board SMD R3 A000073* (Arduino SRL, Strambino, Italy) powered by a 24V power supply via transistor amplifiers (S1 File shows the accompanying Arduino code). The actuation unit and needle are placed inside the MRI bore, while the MR-unsafe components (i.e., the valves, the Arduino Uno board, transistor amplifiers, and power supply) must be placed outside the Faraday cage of the MRI scanner. Therefore, tubes were used to connect the actuation unit to the valves.

## 2.4 Final prototype

Following our previous design [16], the needle in this study consists of six MRI conditional superelastic straight annealed Nitinol rods with a diameter of 0.25 mm and a length of 200 mm (Fig 5A). These needle segments were held together at the tip using 10-mm long *Heat Shrink Tubing 103–0139* (Nordson Medical Corp., Westlake, OH, USA, expanded inner diameter 0.814 mm, wall thickness 0.013 mm). This tube was employed to limit the needle segments from diverging while only minimally increasing the needle diameter. To maintain its position at the needle tip, the heat shrink tube was glued to one of the needle segments using *Pattex Gold Gel 1432562* (Pattex, Henkel AG & Co, Düsseldorf, Germany). The remaining needle

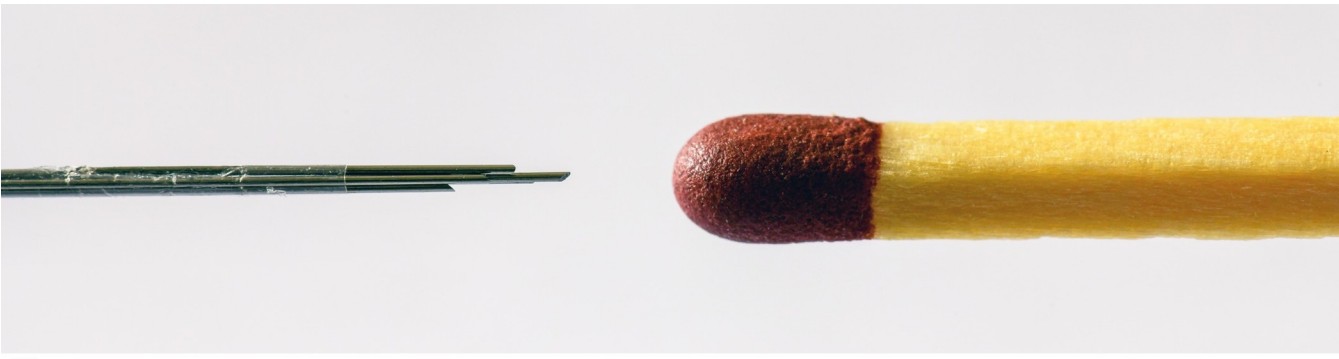

**(A)**

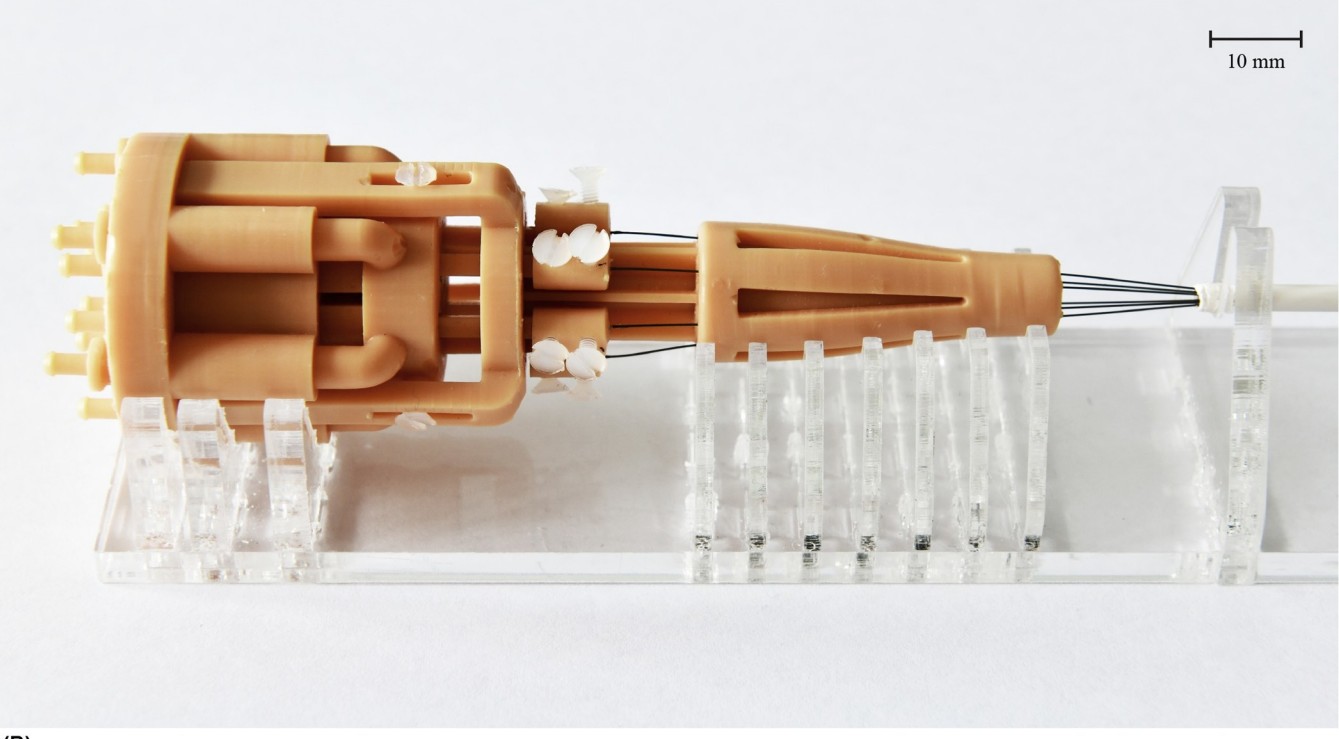

**(B)**

**Fig 5. Pneumatic ovipositor needle.** (A) Close-up of the needle tip consisting of six Nitinol needle segments held together by a heat shrink tube that is glued to one of the six needle segments. (B) Assembled prototype manufactured using the stereolithography technology in Dental Model V2 resin RS-F2-DMBE-02 (Formlabs, Somerville, MA, USA). A transparent PolyMethyl MethAcrylate (PMMA) structure supports the prototype.

segments can move freely back and forth through the heat shrink tube. The resulting total diameter of the needle, including the heat shrink tube, is 0.84 mm.

A stereolithography 3D-printing process was used for the production of the actuation unit prototype due to its ability to produce high-resolution features in the range of 25 μm [56]. All components of the actuation unit were printed using a Formlabs Form 3B printer with a layer height of 0.025 mm using MR safe *Dental Model V2 resin RS-F2-DMBE-02* (Formlabs, Somerville, MA, USA) with the exception of the screws. MR safe polyamide plastic screws of type *Toolcraft 839944 DIN 963 M2 x 10 mm* (Toolcraft Machining, Inc., Germantown, WI, USA) were selected. The assembled actuation unit has a length of 100 mm and an outer diameter of 35 mm (Figs 4B and 5B).

## 3. Performance evaluation

### 3.1 Experiment goal

To evaluate the performance of the Pneumatic Ovipositor Needle under controlled conditions, an experiment in gelatin phantoms was performed. The experiment's goal was twofold: investigate the performance behavior of the needle actuated by the pneumatic actuation unit for (1) different stroke lengths and (2) different piston interval times. The performance of the developed Pneumatic Ovipositor Needle was evaluated in gelatin phantoms in terms of the slip of the needle with respect to the gelatin phantom tissue. More specifically, we calculated the slip ratio of the needle while it advanced through the phantom tissue, using Eq 2.

$$s_{\mathrm{ratio}} = 1 - \left( \frac{d_{\mathrm{m}}}{d_{\mathrm{t}}} \right) \tag{2}$$

Where $d_{\mathrm{m}}$ and $d_{\mathrm{t}}$ are the measured and theoretical maximum traveled distance, respectively. Per cycle, the theoretical maximum traveled distance ($d_{\mathrm{t}}$) equals the stroke length ($S$). To evaluate the slip ratio of the needle throughout all actuation cycles, we used a sensor instead of MRI to continuously measure the traveled distance ($d_{\mathrm{m}}$).

### 3.2 Experiment variables

The dependent variable was the slip ratio ($s_{\mathrm{ratio}}$) between the needle and gelatin for each cycle.

The independent variables were the stroke length ($S$) of the needle segments and the piston interval time ($I$). To investigate the effect of the stroke length ($S$), $S$ was set at 2 or 4 mm, while keeping $I$ at 0.5 s. For investigating the effect of the interval time ($I$), $I$ was set at 0.5, 0.3, or 0.1 s, while keeping $S$ at 4 mm. A shorter interval time means a shorter cycle time, so in theory this results in the needle moving faster through the substrate.

The control variables were the system (gauge) pressure ($P$) of 0.5 bar and the number of actuation cycles ($C$) set to 30. To evaluate the slip ratio ($s_{\mathrm{ratio}}$) of the needle throughout all actuation cycles, the traveled distance ($d_{\mathrm{m}}$) was measured continuously. Table 1 shows the four experiment conditions evaluated in gelatin phantoms.

### 3.3 Experimental setup

Fig 6 shows the experimental setup consisting of the needle connected to the actuation and control units, cart, gelatin phantom, air supply, and data acquisition unit. Instead of moving the needle toward the gelatin, the gelatin was placed on a low-friction aluminum cart and moved toward the actuation unit over PolyMethyl MethAcrylate (PMMA) rails. The principle of needle insertion with zero external push force holds if the self-propelled needle pulls the

**Table 1. Experimental conditions and mean slip ratio mean for the phantom experiment.**

| Condition | Stroke length, $S$ (mm) | Interval time, $I$ (s) | Number of actuation cycles, $C$ | Total theoretical maximum traveled distance, $d_{\mathrm{t}}$ (mm) | Total mean measured traveled distance, $d_{\mathrm{m}}$ (mm) (mean±std) | Slip ratio, $s_{\mathrm{ratio}}$ (mean±std) |
|---|---|---|---|---|---|---|
| S2-I05 | 2 | 0.5 | 30 | 60 | 4.3±3.6 | 0.926±0.123 |
| S4-I05 | 4 | 0.5 | 30 | 120 | 7.7±4.6 | 0.935±0.089 |
| S4-I03 | 4 | 0.3 | 30 | 120 | 10.4±6.2 | 0.912±0.081 |
| S4-I01 | 4 | 0.1 | 30 | 120 | 5.0±2.4 | 0.955±0.049 |

The following information is reported: The condition name, stroke length $S$ (mm), interval time $I$ (s) between pressurization of the subsequent pneumatic cylinders of the actuation unit, number of actuation cycles the needle was actuated for, total theoretical maximum traveled distance $d_{\mathrm{t}}$ (mm) that the gelatin phantom cart would have traveled if no slip at all occurred, total mean measured traveled distance $d_{\mathrm{m}}$ (mm), and slip ratio $s_{\mathrm{ratio}}$.

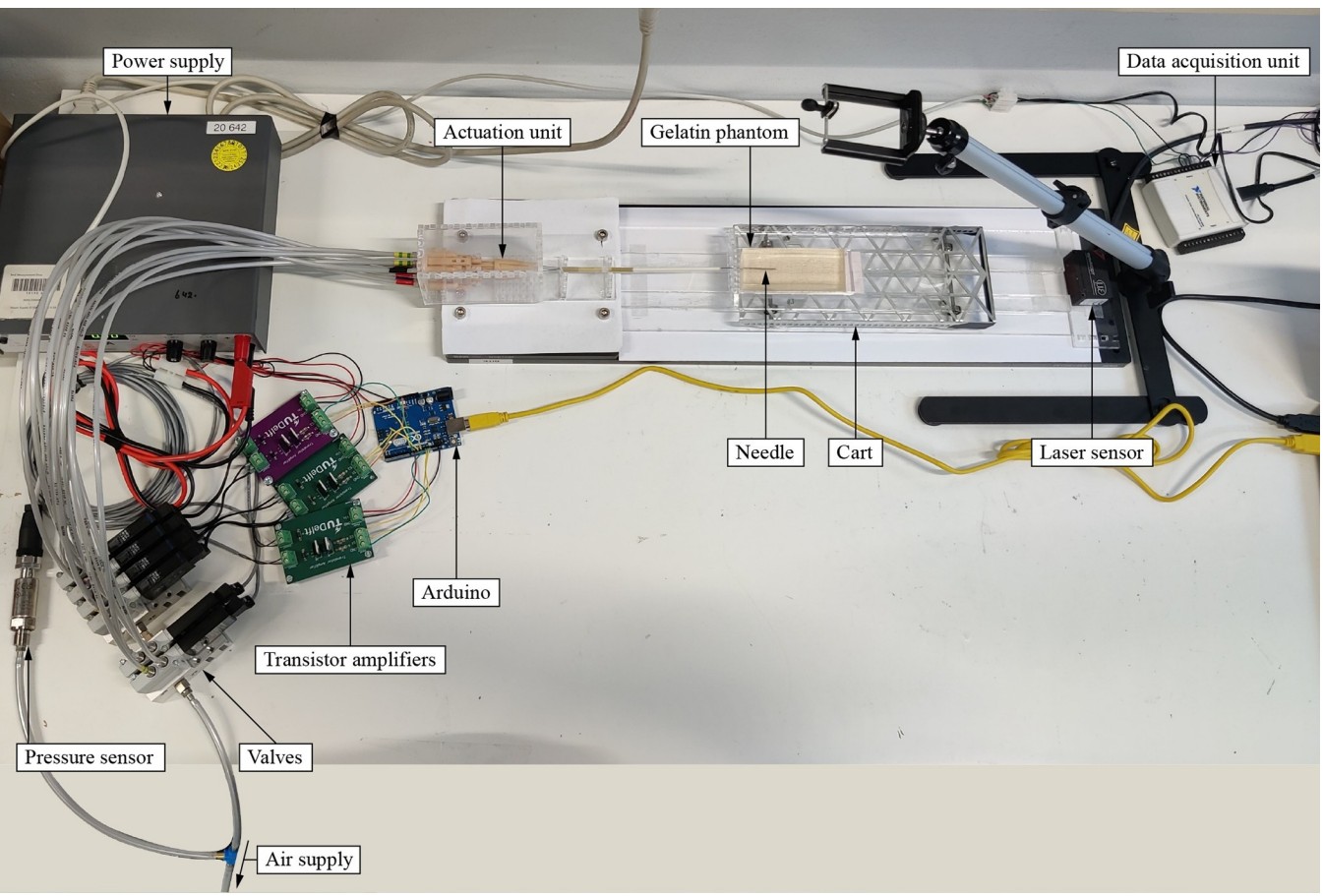

**Fig 6. Photo of the experimental setup used during the phantom experiment, with the needle inserted into the gelatin phantom.**

tissue toward itself by pulling itself deeper into the tissue, thereby pulling the cart toward the needle. The cart (220 x 90 mm$^2$) allowed for low-friction movement using four ball bearings, following the experiment design by Scali *et al.* [51]. Millimeter paper was attached to the cart and used as a reference during needle insertion. A linear laser displacement sensor of type *optoNCDT* (Micro-Epsilon Messtechnik GmbH & Co. KG, Ortenburg, Germany) was used to record the position of the cart during the experiments. A data acquisition unit *NI USB-6211 16-bit* (National Instruments Corporation, Austin, TX, USA) in conjunction with *LabVIEW 2014* (National Instruments Corporation, Austin, TX, USA) was used to collect the laser sensor data. The air pressure entering the valves was monitored using a pressure sensor of type *Autosen AP202* (Autosen GmbH, Essen, Germany) and regulated using a pressure regulator of type *Norgren B72G-2GK-ST3-RMN* (Norgren Ltd, Lichfield, United Kingdom), which was connected to the air supply. To connect the control unit to the actuation unit, we used 30-cm long, 4-mm outer diameter air hoses of type *Festo 152584 PUN-4X0,75-SI* (Festo AG & Co. KG, Esslingen am Neckar, Germany). The air supply and pressure regulator were connected to the control unit using 200-cm long, 6-mm outer diameter air hoses of type *Festo 152586 PUN-6X1-SI* (Festo AG & Co. KG, Esslingen am Neckar, Germany).

For the experiments, gelatin powder of type *Dr. Oetker 1-50-230004* (Dr. Oetker Professional, Amersfoort, The Netherlands) was mixed with water with a gelatin weight ratio (wt) of 10%, resulting in a modulus of elasticity of 17 kPa [51], which is similar to that of prostate

tissue [57,58]. We chose to create gelatin phantoms with a modulus of elasticity similar to that of prostate tissue because there is limited knowledge in the scientific literature about mimicking other mechanical properties of prostate tissue in gelatin phantoms relevant for needle insertion. The gelatin/water mixture was poured into silicone molds and stored overnight at 5°C to solidify. Afterward, the gelatin phantoms were cut to their final dimensions (width 38 mm, length 100 mm, height 20 mm).

## 3.4 Experiment procedure

For each measurement, a new gelatin phantom (mean ± standard deviation = 62.5 ± 4.2 g) was placed on the cart. The needle was inserted over an initial distance of 35 mm inside the gelatin, to ensure initial contact between the needle segments and the gelatin and to ensure the prototype was inserted in a straight direction. The laser sensor was turned on and the needle actuation started. Each test condition was repeated ten times. During every measurement, a linear laser displacement sensor captured the position of the cart (S1 Appendix shows a detailed explanation of the steps in the experiment protocol).

## 3.5 Data analysis

The raw data recorded by the laser sensor during the measurements was imported in MATLAB R2020B (S2 File shows the accompanying MATLAB code and S1 Data the raw data files). The data included the position data of the cart against the time. A Savitzky-Golay filter was used for filtering noise. To evaluate the performance, we calculated the slip ratio ($s_{ratio}$) for each cycle (Eq 2). For the stroke length evaluation, we performed the Wilcoxon signed-rank test for two related groups of non-parametric data (i.e., S2-I05 and S4-I05). For the interval time evaluation, we performed Friedman's ANOVA for three related groups of non-parametric data (i.e., S4-I05, S4-I03, and S4-I01). To further examine the statistical differences between the data, three Wilcoxon signed-rank tests were conducted. The significance level was set at $p < 0.05$.

## 3.6 Results

Fig 7 shows the slip ratio over the number of actuation cycles for each experiment condition. Table 1 shows the mean and standard deviation of the slip ratio for each experiment condition. The mean slip ratios for Conditions S2-I05 and S4-I05 are 0.926 ± 0.123 and 0.935 ± 0.089, respectively. However, the Wilcoxon signed-rank test showed there was no significant difference between both conditions ($z = -0.662$, $p = 0.508$) when comparing the slip ratios for the different stroke lengths. The mean slip ratios for Conditions S4-I05, S4-I03, and S4-I01 are 0.935 ± 0.089, 0.912 ± 0.081, and 0.955 ± 0.049, respectively. A Friedman test ($p = 1.588e-10$) showed that there are at least two conditions with significant differences from each other when comparing the slip ratios for the different interval times. The Wilcoxon signed-rank tests showed that the slip ratio of Condition S4-I01 is significantly different from the slip ratios of Conditions S4-I05 ($z = -2.905$, $p = 0.37e-3$) and S4-I03 ($z = -8.230$, $p = 1.881e-16$). The slip ratio of Condition S4-I05 also proved to be significantly different from the slip ratio of Condition S4-I03 ($z = 3.59$, $p = 3.294e-4$). Furthermore, the slip ratio variability was lower for a shorter stroke length (i.e., 2 mm) compared to a longer stroke length (i.e., 4 mm). Additionally, the slip ratio variability was reduced with lower interval times (i.e., 0.3 and 0.1 s) in contrast to a higher interval time (i.e., 0.5 s). Fig 8 shows typical graphs retrieved by the sensor data for the different experiment conditions. Despite the lowest interval time condition showing the highest slip ratio, the needle moved faster through the substrate because of its shorter cycle time. In

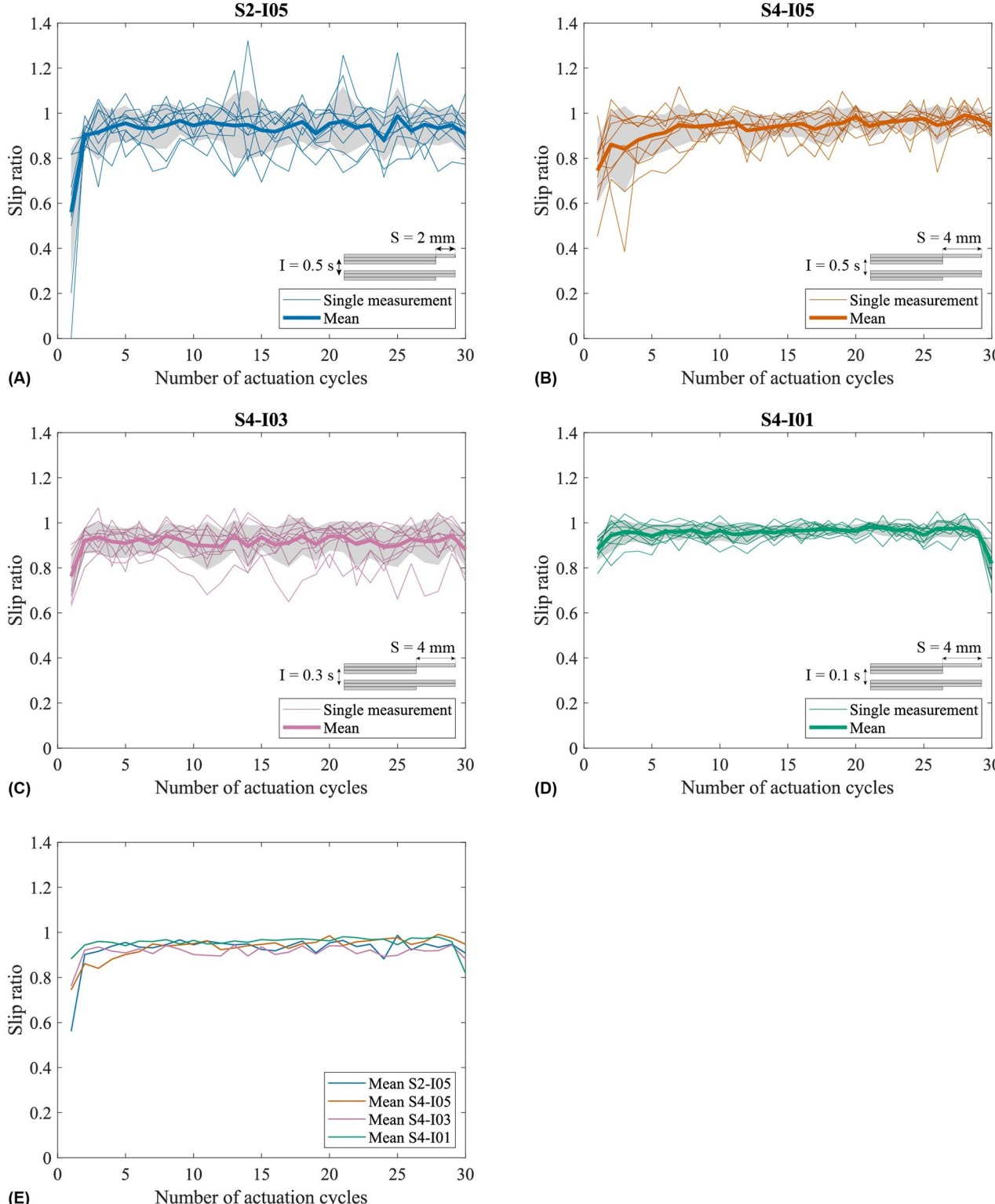

**Fig 7. Results of the phantom experiment.** The subfigures show the slip ratio over the number of actuation cycles of the needle in gelatin phantoms for (A) Pneumatic Ovipositor Needle actuated with a 2-mm stroke length and a 0.5-s interval time, (B) Pneumatic Ovipositor Needle actuated with a 4-mm stroke length and a 0.5-s interval time, (C) Pneumatic Ovipositor Needle actuated with a 4-mm stroke length and a 0.3-s interval time, and (D) Pneumatic Ovipositor Needle actuated with a 4-mm stroke length and a 0.1-s interval time. The thin lines represent the single measurements, the thick line is the mean value, the gray area around the mean value represents the standard deviation. (E) Mean slip ratios for the different experiment conditions over the number of actuation cycles.

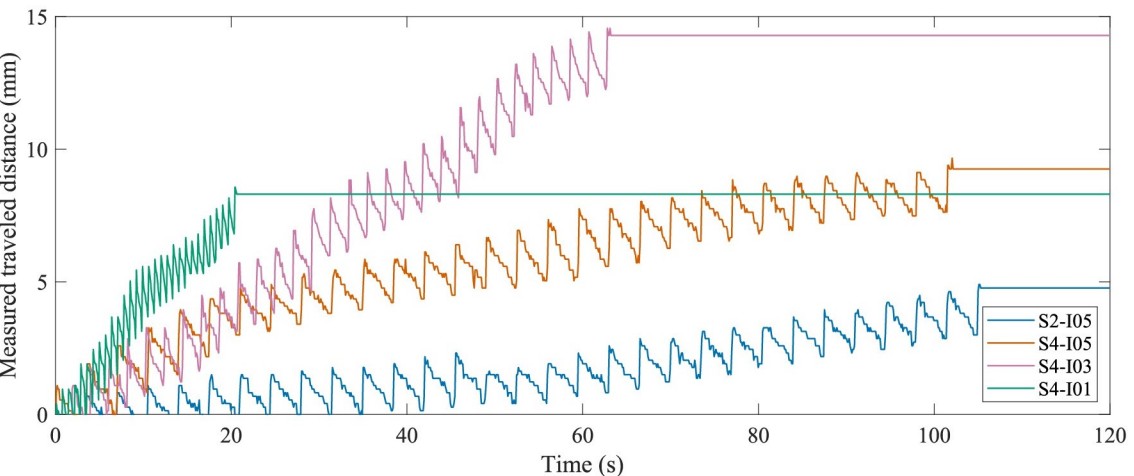

**Fig 8. Plot of measured distance traveled by the cart (mm) vs. time (s) of the phantom experiment, showing typical graphs retrieved by the sensor data for each condition evaluated.** The local minima show the position of the cart when all needle segments are advanced, and the local maxima represent the position of the cart when all needle segments are retracted during the step-by-step actuation. The distance between two peaks defines a cycle. For all conditions, the needle was actuated for 30 actuation cycles, so for 0.5-s, 0.3-s, and 0.1-s interval times, the measurement took 105 s, 63 s, and 21 s, respectively. A shorter interval time means a shorter cycle time, so this resulted in the needle moving faster through the substrate.

S1 Video of this article, a video shows the tip of the Pneumatic Ovipositor Needle in gelatin during actuation.

## 3.7 Additional MRI experiment

To demonstrate the potential of the needle inside a real MRI environment, an additional proof-of-principle experiment using *ex vivo* porcine liver tissue in a preclinical 7-Tesla MRI scanner (MR Solutions, Guildford, United Kingdom) at the Amsterdam University Medical Center (AUMC, department of Biomedical Engineering and Physics) was performed. Due to the limited accessibility of *ex vivo* human prostate tissue, we employed *ex vivo* porcine liver tissue as an alternative, given its comparable modulus of elasticity of 15–20 kPa [59] to prostate tissue [57,58]. Ethical approval was not required for the study involving animals in accordance with the local legislation and institutional requirements because the tissue sample used was commercially available porcine liver tissue obtained from a local butcher.

Fig 9 shows the experimental setup consisting of the needle connected to the actuation and control units, the MRI scanner, the tissue sample in a tissue box on a low-friction design located inside a Radio Frequency (RF) coil, and nitrogen supply. For a more comprehensive explanation of the low-friction design inside the RF coil, please refer to the description provided in our previous work [16].

We prepared the biological sample (width 50 mm, length 90 mm, height 10 mm, weight 46 g) by placing a piece of *ex vivo* porcine liver tissue (width 50 mm, length 45 mm, height 10 mm) in the tissue box with liquid agar (1.0 wt%). The liver tissue was positioned at the distal end of the tissue box and the remaining part of the tissue box was filled with agar, which facilitated the insertion of the needle for approximately 45 mm into the agar before reaching the liver tissue. Storing the box in the refrigerator overnight ensured fixation of the tissue in the agar.

A 3D gradient-echo acquisition was performed continuously for 3 minutes to image the needle position during needle actuation with respect to the liver tissue and the tissue box.

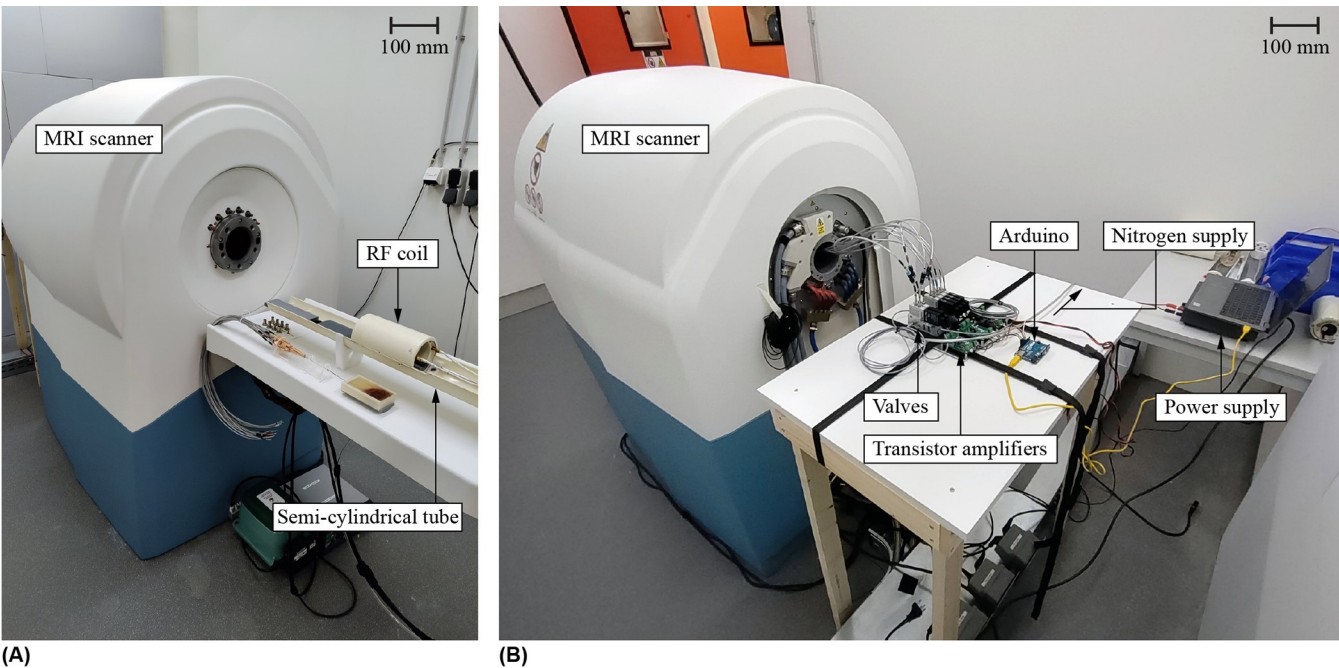

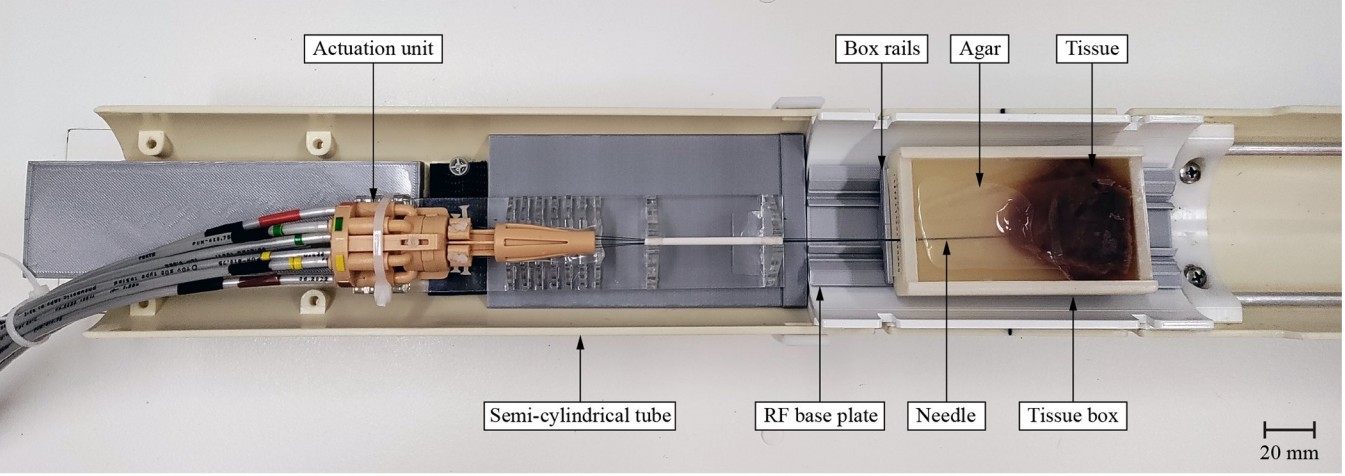

**Fig 9. Experimental setup used during the Magnetic Resonance Imaging (MRI) experiment.** (A) Front of the MRI scanner where the Pneumatic Ovipositor Needle was placed in a semi-cylindrical tube that slides into the MRI bore. The semi-cylindrical tube supports a Radio Frequency (RF) coil for signal reception, in which the tissue box was placed. (B) Back of the MRI scanner where the hoses connecting the valves to the Pneumatic Ovipositor Needle and the actuation unit enter the MRI bore. (C) Close-up of the semi-cylindrical tube from above with the actuation unit and the tissue box on the RF base plate, guided on box rails. In the tissue box, the *ex vivo* porcine liver tissue was embedded in solidified 1-wt% agar.

Because we used a continuous pseudo-radial k-space sampling pattern, this allowed for retrospective 3D image reconstruction at different temporal resolutions. Therefore, we obtained high-quality 3D images of both the initial and final static position of the needle, as well as reconstructions with higher temporal resolution (7.5 s) during movement of the tissue along the needle (see S2 Appendix for imaging parameters and S3 Appendix for a detailed explanation of the steps in the experiment protocol). Fig 10 shows the MR images of the needle tip positions and S2 Video shows the dynamic MRI video. The measurement showed that the needle was able to propel itself forward inside the liver tissue with a slip ratio of 0.88.

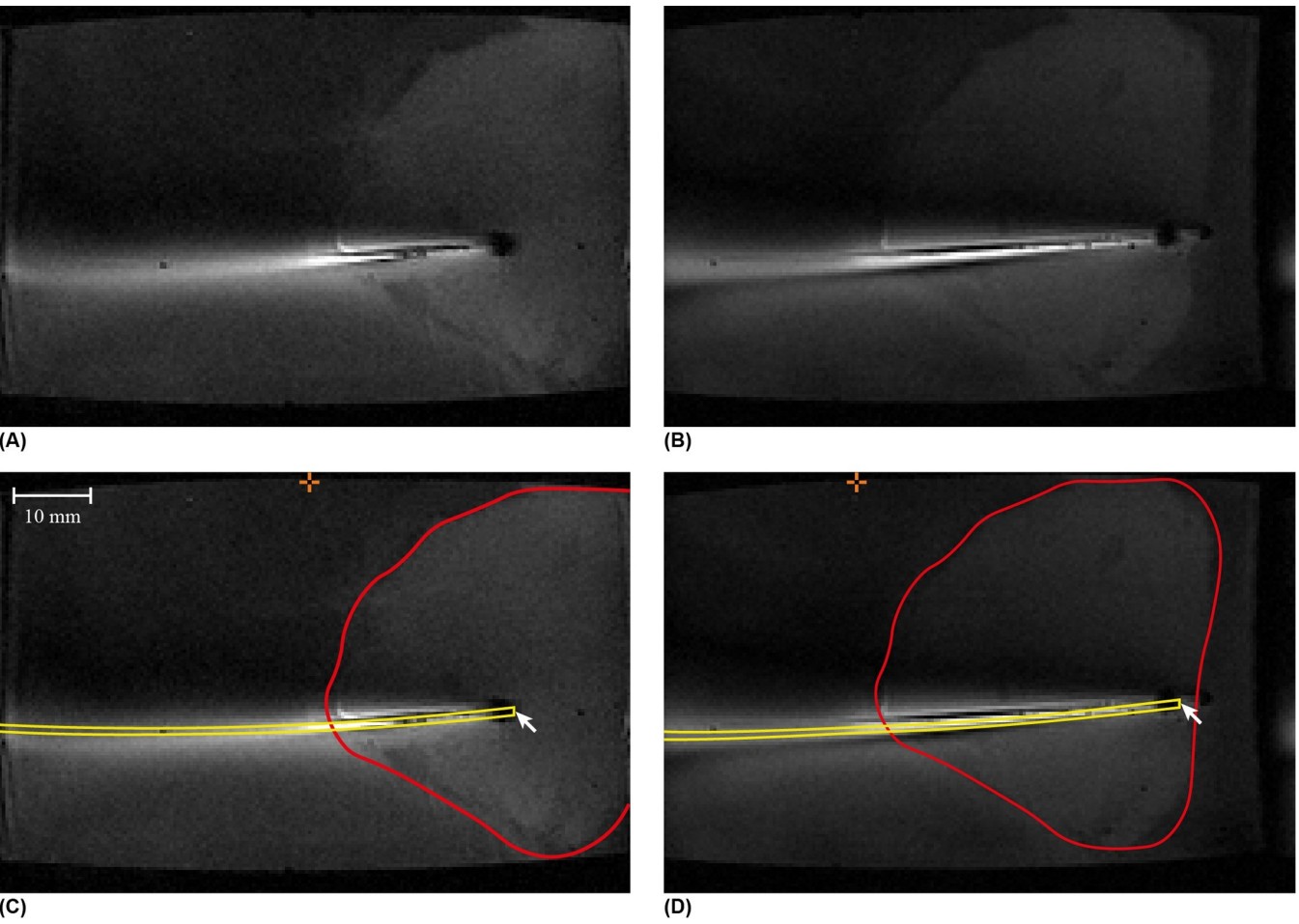

**Fig 10. Magnetic Resonance (MR) images of the needle inside the agar and *ex vivo* porcine liver tissue.** Original MR images (A) where the needle tip is inserted through the agar and partially through the liver tissue and (B) after actuation over thirty actuation cycles. Annotated MR images (C) where the needle tip is inserted through the agar and partially through the liver tissue and (D) after actuation over thirty actuation cycles. The yellow and red contours show the needle and liver tissue, respectively. The arrow marks the needle tip. The orange crosshair shows the reference point on the side of the box that indicates a 40-mm insertion depth.

## 4. Discussion

### 4.1 Main findings

Compared to previous work done by Scali *et al*. [51], our measured mean slip ratio is high. A possible explanation for our high slip ratio results could be that the insertion force required to move the needle segments forward was high because of the high insertion speed caused by the pneumatic actuation. DiMaio and Salcudean [60] and Meltsner *et al*. [61] showed that both cutting and friction forces of needles in tissue phantoms increase with increasing insertion velocities. Furthermore, the compressibility of air may have resulted in a lower actual stroke length compared to the stroke length the needle was actuated for.

### 4.2 Limitations

In both phantom and MRI experiments, the inertia of the cart, as well as the experienced friction might have influenced the slip results. The mass of the cart and the gelatin phantom or tissue tends to increase the slip in the retracting phase of the needle segments, whereas the

contrary holds for the advancing phase. In the retracting phase, the resultant force on the needle inside the tissue (phantom) is likely to be lower than the force required to overcome the bearing friction and cart inertia, therefore an effect of inertia and bearing friction cannot be ruled out.

In the experimental setup, air hoses were used to interconnect the air supply with the control and actuation units. However, in the MRI experiment, the length of the air hoses connecting the control and actuation unit was required to be longer than during the phantom experiment (70 cm vs. 30 cm, respectively) to maintain a certain distance between the MR unsafe control unit and the center of the MRI scanner. Assuming a constant nominal standard airflow for the pneumatic valves, an increase in the air hose length results in an increase in the pressure drop and delay and friction effects in the air hose [27]. These effects cause a variation in the pressure inside the actuation unit between the phantom experiment with shorter air hoses and the MRI experiment with longer air hoses, potentially affecting the actual stroke length of the needle segments and slip ratio of the needle. Furthermore, to prevent air leakage in future versions of the Pneumatic Ovipositor Needle, replacing the pistons by bellows could be investigated [62,63].

In the experiments, gelatin phantoms and *ex vivo* porcine liver tissue with a modulus of elasticity similar to that of prostate tissue were used. However, for needle insertion experiments, other mechanical properties, such as the needle-tissue friction coefficient, shear modulus, and ultimate strength of the tissue are of higher interest than the modulus of elasticity of the tissue. Furthermore, material properties may vary from human to porcine tissue [64]. Therefore, we recommend investigating the mechanical properties of human prostate tissue, *ex vivo* tissue specimens, and tissue-mimicking materials that are important for needle insertion and propulsion, thereby also considering the tissue's nonlinearity and heterogeneity.

## 4.3 Recommendations and future research

The Pneumatic Ovipositor Needle was designed as a delivery needle for TPLA to treat prostate cancer. The integration of the optical fiber in the Pneumatic Ovipositor Needle requires a hollow core in the actuation unit positioned between the pneumatic cylinders, allowing six needle segments, concentrically arranged around a 300-μm diameter optical fiber for TPLA [65], resulting in a total needle outer diameter of 0.8 mm. In the two-phase motion sequence of the needle, retracting all six needle segments simultaneously could advance the centrally positioned optical fiber into the substrate. The surface area of the needle segments in direct contact with the optical fiber is the same as the surface area of the needle segments in direct contact with each other in the current prototype. Consequently, we do not expect the optical fiber to influence the friction forces of the needle. However, in theory, the optical fiber does increase the cutting force during the retraction phase. In future work, it will be interesting to perform experiments to determine the cutting and friction forces acting on the needle.

To develop a full picture of the Pneumatic Ovipositor Needle in a clinical setting, *in vivo* experiments are needed. When moving toward an *in vivo* study, we foresee some challenges, including the presence of fluids (i.e., blood) and multi-layered tissue. The presence of fluids like blood may reduce the needle-tissue friction required for the self-propelled motion. On the other hand, the parasitic wasp is able to move effectively through fluid-like substances like fruits thanks to directional friction patterns that increase friction [66]. Inspired by these friction patterns, Parittotokkaporn *et al.* [67] added a directional friction pattern to the needle surface. This could be an interesting avenue to explore for our needle in the future.

Another challenge that arises from an *in vivo* model is the presence of multiple tissue layers between the insertion point, which is the perineum, and the target position within the prostate

gland. Each tissue layer adds its own cutting, stiffness, and friction forces [68,69]. The ability of the self-propelled needle to advance in multi-layered tissue-mimicking phantoms with varying stiffness has been exemplified successfully in a previous study [51]. In addition, real tissue is inhomogeneous and differences in tissue characteristics are present within and between human beings, which makes the forces that act on the needle inside the body difficult to predict.

Lastly, during the performance evaluation in this study, we kept the actuation unit stationary while the tissue was placed on a low-friction cart that moved toward the needle. In clinical practice, the patient will have to remain stationary while the needle self-propels inside the tissue. To accommodate this, the needle could be moved inside the actuation unit to manipulate the needle towards the patient, following the pace of the self-propelled motion of the needle. To further increase the effectiveness of the procedure and decrease the chance of unwanted tissue damage [32] and pubic arch interference [70], the ability to actively and intuitively steer the tip of the needle during the procedure should be researched.

## 5. Conclusion

In this study, a pneumatic actuation unit for a self-propelled ovipositor-inspired needle consisting of six needle segments for MRI-guided procedures is presented. The pneumatic actuation unit design allows an adjustable output stroke length of 2 to 10 mm, where the stroke length is the distance that each needle segment travels per needle actuation cycle. Furthermore, the actuation unit and needle consist solely of MR safe/conditional materials and the control unit allows for control of the system with an adjustable interval time and a single air input. The evaluation of the prototype in 10-wt% gelatin phantoms showed that the needle was able to self-propel through the phantoms. Additionally, we measured the lowest slip ratio of 0.912 ±0.081 for a stroke length of 4 mm and an interval time of 0.3 s. The experiment of the prototype inside a preclinical MRI scanner showed the needle could also self-propel through *ex vivo* porcine liver tissue with a slip ratio of approximately 0.88. The pneumatic actuation unit is a step forward in developing a self-propelled needle for MRI-guided percutaneous procedures.

## Supporting information

**S1 File. Arduino code to control Arduino board in phantom and MRI experiments.** S1 File Arduino contains the Arduino program code for the phantom and MRI experiments in this study. This code can be run in the Arduino IDE (Integrated Development Environment) software by uploading it to the Arduino Uno board. The Arduino IDE software can be downloaded on the Arduino website: https://www.arduino.cc/en/software.
(TXT)

**S2 File. MATLAB software to run the phantom experiment data analysis.** Running the analysis S2 File SlipPneumaticOvipositorNeedle requires the supplementary raw data set of the phantom experiment (i.e., S1 Data). Therefore, in the MATLAB code, you need to replace 'C:\path\your\data\folder\' with the actual path to your downloaded S1 Data folder.
(M)

**S1 Appendix. Phantom experiment protocol.** S1 Appendix contains the experimental procedure for each measurement inside the gelatin phantoms.
(TXT)

**S2 Appendix. Magnetic resonance parameters.** S2 Appendix contains the Magnetic Resonance parameters set for the MRI experiment in this study.
(TXT)

**S3 Appendix. MRI experiment protocol.** S3 Appendix contains the experimental procedure for the measurement inside *ex vivo* porcine liver tissue.
(TXT)

**S1 Data. Raw data set of the phantom experiment.** The S1 Data file contains the raw data sets of the phantom experiment.
(ZIP)

**S1 Video. Video of the needle moving through gelatin.** S1 Video Phantom contains a video of the actuation unit and needle tip in the gelatin phantom for Condition S4-I01.
(MP4)

**S2 Video. MRI video of the needle moving through tissue.** S2 Video MRI contains the dynamic MRI video of the experiment in *ex vivo* porcine liver tissue.
(MP4)

## Acknowledgments

We would like to thank Gertjan Mulder and Jacques Brenkman for discussions about pneumatics and their help in designing the setup, and we would like to thank Judith de Vos for preparing the tissue sample.

## Author Contributions

**Conceptualization:** Jette Bloemberg, Bruce Hoppener, Aimée Sakes, Paul Breedveld.

**Data curation:** Jette Bloemberg, Bruce Hoppener.

**Formal analysis:** Jette Bloemberg, Bruce Hoppener.

**Funding acquisition:** Paul Breedveld.

**Investigation:** Jette Bloemberg, Bruce Hoppener, Bram Coolen.

**Methodology:** Jette Bloemberg, Bruce Hoppener, Aimée Sakes, Paul Breedveld.

**Project administration:** Paul Breedveld.

**Resources:** Jette Bloemberg, Bram Coolen.

**Software:** Jette Bloemberg, Bruce Hoppener.

**Supervision:** Aimée Sakes, Paul Breedveld.

**Validation:** Jette Bloemberg, Bruce Hoppener, Bram Coolen.

**Visualization:** Jette Bloemberg.

**Writing – original draft:** Jette Bloemberg.

**Writing – review & editing:** Jette Bloemberg, Bruce Hoppener, Bram Coolen, Aimée Sakes, Paul Breedveld.

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
