## [Decision Letter · Decision Letter 0]

22 Jan 2024

PONE-D-23-38960Design and evaluation of a pneumatic actuation unit for a wasp-inspired self-propelled needlePLOS ONE

Dear Dr. Bloemberg,

Thank you for submitting your manuscript to PLOS ONE. After careful consideration, we feel that it has merit but does not fully meet PLOS ONE’s publication criteria as it currently stands. Therefore, we invite you to submit a revised version of the manuscript that addresses the points raised during the review process.

The authors are recommeded to revise the manuscript in a more comprehensive way by emphasising the main focus(es) of the work, according to the reviewers' comments. Please carefully address each point that has been pointed out by the reviewers.

We look forward to receiving your revised manuscript.

Kind regards,

Baeckkyoung Sung, Ph.D.

Academic Editor

PLOS ONE

Journal Requirements:

5. We are unable to open your Supporting Information file [S1 Software.Arduino and S3 Software. MATLAB]. Please kindly revise as necessary and re-upload.

Reviewers' comments:

Reviewer's Responses to Questions

**Comments to the Author**

1. Is the manuscript technically sound, and do the data support the conclusions?

Reviewer #1: Yes

Reviewer #2: Yes

Reviewer #3: Yes

2. Has the statistical analysis been performed appropriately and rigorously? 

Reviewer #1: Yes

Reviewer #2: Yes

Reviewer #3: Yes

3. Have the authors made all data underlying the findings in their manuscript fully available?

Reviewer #1: No

Reviewer #2: Yes

Reviewer #3: Yes

4. Is the manuscript presented in an intelligible fashion and written in standard English?

Reviewer #1: Yes

Reviewer #2: Yes

Reviewer #3: Yes

5. Review Comments to the Author

Reviewer #1: Thanks very much for this interesting work you are presenting. While being good to read, it lacks focus (is it the needle arrangement? is the pneumatic actuation? Is it MRI compatibility? Is it self-propelling insertion into liver?) and is lengthy in some parts (especially in the introduction), while you are citing a lot of other projects, it turns out that these are mostly from a group around Baena, by Scali or by your own group. A broader review is definitely necessary.

The theoretic background should either be more comprehensive (preferable) or left out A single equation describing what a line of text said before is not useful. It would be interesting to know, how the difference of static and dynamic friction together with the stiffness of the needles, the stiffness of the tissue and – most important! – the strength of the tissue affect the movement. Also, the steerability (or straightness) of the needle’s movement must be assessed. Where could there be potential for improvement? How must the tips of the single needles be cut? How does the friction between the needles and the core (the laser fibre) affect its performance? How do fluids like blood (clotting?) or fat affect the performance in theory?

If MRI-compatibility is important, I wonder how a 30 cm tube between your system and the valve block (which is electromagnetic) is MRI-compatible. But if you use longer tubes (like those 7 m between the nitrogen (why nitrogen, when you talk about air?) you will lose a lot of dynamics in the system. This should also be addressed and discussed.

Detailed issues

L40 type Trans blank Perineal

L81 These refs are all from the same group. Is there nobody else on the planet working on these bio-inspired needles?

L103 This equation is just the beginning: how are F_cut and F_friction influenced?

L113 If other people have done that before, what is so new about your needle? � find good focus

L126 If electric motors are the problem for MRI, why not look a refs of people who used pneumatics? � it’s a question of focus the refs are all from the same group

L195 Tube length limits actuation frequency

L202 Quantify “minimally increasing” How thick is the shrink tube?

L214 what pressure are you envisioning? What frequency of actuation? How do you deal with leakage and noise?

L222 please avoid first-person sentences and use passive voice.

L233 what friction coefficients are you expecting?

L241 here, you use 0.5 bar in L346 you talk about 3 bar. What is true or why is it different?

L258 If the experimental setup dates back to 2019 what has happened ever since?

L267 in my opinion, 300 m distance between needle and valvle block are far too short for MRI compatibility!

L277 young’s modulus is of inferior interest here. The strength of the material and the friction are of most importance!

L334 a demonstration is nice but should be described much shorter. As an alternative this could be a real experiment where you show how straight your needle goes, how little the image distortions are how fast you go etc. But not like “see, it works!”

L340 again: young’s miodulus is not the point, strength and friction are more important.

L361 if you refer to your previous work, make clear what is new about this here

L363 do you use air or nitrogen?? Please be precise!

L368 Liver has a density around 1.079 g/cm³ why is your 45 cm³ piece so heavy?

L415 The discussion is far too long. Please focus on your main research question and do not discuss future projects. This is a paper not a Master’s thesis!

L559 55 ref are more than enough…

Fig 1 give the reader an impression of size do not rely on colours

Fig 2 also misses a scale

Fig 5 b misses a scale

Fig 9 a misses a scale

Fig 10 b misses a scale

Reviewer #2: The paper proposes a novel self-propelled needle designed for MRI-guided percutaneous laser ablation of the prostate. The self-propelling mechanism is inspired by the wasp’s ovipositor and consists of six needle segments that are driven by pneumatic actuators independently. The newly developed needle was tested in a gel phantom and an ex vivo tissue. While the mechanism has been proposed previously, this work appears to be the first attempt to make such a mechanism work in the MRI environment.

The work offers a solid contribution to the research community. If successful, the proposed mechanism will allow placing a needle into the prostate under live MRI guidance more efficiently, as it addresses the issues associated with MRI guidance, such as the narrow workspace in the MRI gantry and the deviation of the needle due to tissue-needle interaction. The paper is well written with sufficient detail of the work.

However, the performance of the self-propelling mechanism, especially the slip ratio, is somewhat unconvincing, probably due to the lack of comprehensive understanding of the forces involved in the propelling. This reviewer still thinks that the experimental data are worth publishing, though the paper needs to address the following questions.

(1) (Major) Equation (1) only considers the cutting force and the friction between the needle shaft and the tissue, though it is not clear if they are the dominant forces when considering the propulsion of the needle. If the friction is dominant, the slip ratio in Figure 7 should have increased as the needle advances into the tissue because the friction should be proportional to the contact area (i.e., the length of the needle inside the tissue). However, the figures do not seem to show such an effect except S4-105.

(2) Related to comment 1, did the authors measure the friction and cutting forces independently? They could have measured them by inserting and retracting the needle in the phantom while measuring the force along the needle. (Assuming that the thrust force required for insertion would equal the sum of the friction and the cutting force, while that for retraction would only equal the friction.)

(3) Again related to comment 1, the authors listed the inertia and friction of the cart as limitations in section 5.2. If the slip ratio is not clearly influenced by the insertion length, wouldn’t it be more reasonable to consider those factors in Equation (1)? They are independent of the insertion length. If the authors measured the displacement of the cart, they could have estimated the force from the inertia based on the mass and the acceleration of the cart.

(4) (Moderate) The authors only consider the material property of the prostate, but the prostate tissue might be a small portion of the overall insertion path, probably ~20-30mm out of ~100mm insertion in the transperineal insertion. The rest consists of tissues with significantly different properties, such as muscles, skin, etc. The presented experiment is a good first step, but it would be helpful if the authors discussed what the results mean for other types of tissues (e.g., whether harder/softer tissue results in higher/lower friction ratio).

(5) (Moderate) The authors discuss air leakage as one of the limitations. Have they considered diaphragms or bellows in place of pistons? There were studies used bellows for actuation, such as: Comber DB, IEEE Trans Robot. 2016 Feb;32(1):138-149. doi: 10.1109/TRO.2015.2504981. Epub 2016 Jan 19. PMID: 31105476; PMCID: PMC6519471.

(6) (Minor) Is the dimensions of the needle large enough to accommodate the optic fiber for laser ablation? What is the authors’ plan to scale up this prototype to incorporate the ablation device?

Reviewer #3: It is a well-written paper and I suggest the paper to be accepted with a revision. I think the discussion on the setup for the clinical setting needs to be presented in more details. I understand the limitation of the in-lab experiment however I would like to see further insights on how to do the actuation in a clinical setting. A different actuation method may completely result in a different conclusion. Could a rough experiment be performed?

6. PLOS authors have the option to publish the peer review history of their article (what does this mean?). If published, this will include your full peer review and any attached files.

Reviewer #1: No

Reviewer #2: No

Reviewer #3: No

---

## [Author Response · Author response to Decision Letter 0]

22 Feb 2024

Dear Baeckkyoung Sung, Ph.D., 

Thank you for your time and consideration of our manuscript. Based on the reviewers’ comments below and your editorial comments, we have revised the manuscript accordingly. 

Regarding the points described in your email, we confirm that (1) we adapted the manuscript to meet PLOS ONE’s style requirements, (2) we made all code available without restrictions, (3) we provided the grant numbers for our study in the ‘Funding Information’ section in the submission system and we added an updated financial disclosure statement in our cover letter, (4) we adjusted our ethics statement in Section 3.7 (Lines 368-370), and (5) we re-uploaded Supporting Information files [S1_Software_Arduino.txt and S3_SlipPneumaticOvipositorNeedle.txt] as text files.

Please find our detailed responses to the reviewers below. In the revised manuscript, the changes are indicated using track changes.

Review Comments to the Author

Reviewer #1: Thanks very much for this interesting work you are presenting. While being good to read, it lacks focus (is it the needle arrangement? is the pneumatic actuation? Is it MRI compatibility? Is it self-propelling insertion into liver?) and is lengthy in some parts (especially in the introduction), while you are citing a lot of other projects, it turns out that these are mostly from a group around Baena, by Scali or by your own group. A broader review is definitely necessary.

R1. Thank you for your compliments on the novelty of our work. Our pneumatic actuation unit works in unison with our developed wasp-inspired needle prototype, is MRI compatible, and can thus be used for a wide variety of medical applications. The focus is on the pneumatic actuation unit (see Lines 157-160). 

Unfortunately, to our knowledge, the only groups working on wasp-inspired needles for friction manipulation to reduce tissue damage are Baena’s group at the Imperial College London and Breedveld’s group at the TU Delft. In previous work, we reviewed bio-inspired medical needles, which show other sources of bio-inspiration for percutaneous needles focused on different working principles, such as mosquito-inspired needles (Fung-A-Jou et al., 2023) (Lines 81-82). Furthermore, we shortened the introduction section (Lines 88-92).

Fung-A-Jou Z, Bloemberg J, Breedveld P. Bioinspired medical needles: a review of the scientific literature. Bioinspiration & Biomimetics. 2023. doi: 10.1088/1748-3190/acd905.

The theoretic background should either be more comprehensive (preferable) or left out A single equation describing what a line of text said before is not useful. It would be interesting to know, how the difference of static and dynamic friction together with the stiffness of the needles, the stiffness of the tissue and – most important! – the strength of the tissue affect the movement. Also, the steerability (or straightness) of the needle’s movement must be assessed. Where could there be potential for improvement? How must the tips of the single needles be cut? How does the friction between the needles and the core (the laser fibre) affect its performance? How do fluids like blood (clotting?) or fat affect the performance in theory?

R2. We want to thank the reviewer for these useful suggestions. We, therefore, now elaborate on the in-text explanation of the different forces acting on the needle during insertion into tissue (Lines 100-141). In the explanation, we differentiated between the surface stiffness force in the pre-puncture phase and the cutting and friction forces in the post-puncture phase. Furthermore, we added information on how these different forces arise and by which factors they are affected. The self-propelled motion requires surrounding tissue. Thus, it works in the post-puncture phase where the surface stiffness force is equal to zero when assuming constant mechanical tissue properties in homogeneous tissue. In addition, as the needle propels deeper into the tissue, the surface area of the needle segments in direct contact with the tissue increases, increasing the friction forces on the needle segment linearly with insertion depth while the cutting force remains roughly constant throughout the insertion.

In the discussion in Lines 614-624, we elaborate on the integration of steering in the wasp-inspired needle to increase the effectiveness of the procedure and decrease the chance of unwanted tissue damage and pubic arch interference.

Secondly, we have added an explanation of the effect of integrating an optical fiber into the needle bundle's center on the needle's friction and cutting forces (Lines 575-587). We explain that in the two-phase motion sequence of the needle, retracting all six needle segments simultaneously advances the centrally positioned optical fiber into the substrate. The surface area of the needle segments in direct contact with the optical fiber is the same as the surface area of the needle segments in direct contact with each other in the current prototype. Consequently, we do not expect the optical fiber to influence the friction forces of the needle. However, the optical fiber does increase the cutting force during the retraction phase.

Thirdly, we added a paragraph discussing the foreseen challenges in moving towards in vivo testing (Lines 589-613). In this new paragraph, we discuss the effects of fluids like blood, which may reduce the needle-tissue friction required for the self-propelled motion. On the other hand, the parasitic wasp can move effectively through fluid-like substances like fruits thanks to directional friction patterns that increase friction. Inspired by these friction patterns, Parittotokkaporn et al. (2010) added a directional friction pattern to the needle surface. This could be an interesting avenue to explore for our needle in the future. 

Parittotokkaporn T, Frasson L, Schneider A, Davies B, Degenaar P, y Baena FR. Insertion experiments of a biologically inspired microtextured and multi-part probe based on reciprocal motion. 2010 Annual International Conference of the IEEE Engineering in Medicine and Biology; 2010: IEEE. doi: 10.1109/IEMBS.2010.5627410.

If MRI-compatibility is important, I wonder how a 30 cm tube between your system and the valve block (which is electromagnetic) is MRI-compatible. But if you use longer tubes (like those 7 m between the nitrogen (why nitrogen, when you talk about air?) you will lose a lot of dynamics in the system. This should also be addressed and discussed.

R3. You are correct in that the valve block is currently not MRI-compatible. This is the reason why this part of the test set-up was strapped to a table. In the future, we intend to manufacture a pneumatic actuation unit with pneumatic valves and investigate fluidic logic, to prevent using electronics. Furthermore, we changed “air” in Line 381 to “nitrogen.” 

Detailed issues

L40 type Trans blank Perineal

R4. Thank you for this suggestion. In Line 40, we have now changed TransPerineal to Trans Perineal.

L81 These refs are all from the same group. Is there nobody else on the planet working on these bio-inspired needles?

R5. Please refer to R1. 

L103 This equation is just the beginning: how are F_cut and F_friction influenced?

R6. Please refer to R2. 

L113 If other people have done that before, what is so new about your needle? � find good focus

R7. The novelty of our needle lies in the pneumatic actuation unit, please see Section 1.4. 

L126 If electric motors are the problem for MRI, why not look a refs of people who used pneumatics? � it’s a question of focus the refs are all from the same group

R8. In our goal description, we describe current prototypes of wasp-inspired self-propelled needles. Wasp-inspired self-propelled needles using pneumatics do not exist yet in the scientific literature. Pneumatic actuation systems for general needles exist; they are mentioned in Lines 69-74. 

L195 Tube length limits actuation frequency

R9. The effect of tube length on the airflow is discussed in Section 4.2, Lines 538-546. 

L202 Quantify “minimally increasing” How thick is the shrink tube?

R10. Thank you for pointing out this missing information. In Section 2.4 in Line 228, we added the expanded inner diameter and wall thickness dimensions. 

L214 what pressure are you envisioning? What frequency of actuation? How do you deal with leakage and noise?

R11. In the future, we would like to research the effect of different frequencies on the efficiency of the system. In the discussion, we added a line about investigating the implementation of bellows instead of pistons to prevent air leakage in the system (Lines 529-530). With future implementation, the effect of noise could also be investigated. 

L222 please avoid first-person sentences and use passive voice.

R12. In Sections 3.1 and 3.7, Lines 249-250 and 362-366 were rewritten in passive voice. 

L233 what friction coefficients are you expecting?

R13. The friction coefficient depends on the exact tissue properties in which we test the device. At the moment, we have not performed any experiments to measure this friction coefficient. In the discussion in Lines 594-596, we added that directional friction patterns, as investigated by Parittotokkaporn et al. (2010), could be an interesting avenue to explore.

Parittotokkaporn T, Frasson L, Schneider A, Davies B, Degenaar P, y Baena FR. Insertion experiments of a biologically inspired microtextured and multi-part probe based on reciprocal motion. 2010 Annual International Conference of the IEEE Engineering in Medicine and Biology; 2010: IEEE. doi: 10.1109/IEMBS.2010.5627410.

L241 here, you use 0.5 bar in L346 you talk about 3 bar. What is true or why is it different?

R14. Based on the experiments in gelatin samples we decided to increase the pressure for the ex vivo tissue experiments to improve the cutting capability of the needle. 

L258 If the experimental setup dates back to 2019 what has happened ever since?

R15. It is true that the experimental setup used in this study is similar to the setup described in 2019. However, the design, as explained in Section 2, evaluated with the experimental setup has changed. 

L267 in my opinion, 300 m distance between needle and valvle block are far too short for MRI compatibility!

R16. Please refer to R3. MRI compatibility in this research mainly involves the actuation unit parts and other parts crucial to the experimental setup. 

L277 young’s modulus is of inferior interest here. The strength of the material and the friction are of most importance!

R17. For the working principle of the wasp-inspired needle, the friction coefficient and cutting force are most important during propulsion. However, because we wanted the gelatin phantoms to mimic mechanical tissue properties, we indicated Young’s modulus of the gelatin phantoms. 

L334 a demonstration is nice but should be described much shorter. As an alternative this could be a real experiment where you show how straight your needle goes, how little the image distortions are how fast you go etc. But not like “see, it works!”

R18. Thank you for your suggestion. We shortened the text about the additional MRI experiment and removed it as a separate Section (i.e., Section 4. MRI experiment). Instead, we added it as a subsection to Section 3. Performance evaluation. The new subsection is called “3.7 Additional MRI experiment”. 

L340 again: young’s miodulus is not the point, strength and friction are more important.

R19. Please refer to R17. 

L361 if you refer to your previous work, make clear what is new about this here

R20. Solely the low-friction design inside the RF coil of the pre-clinical MRI scanner was the same as the low-friction design used in our previous work. We also clarified this in the manuscript in Lines 390-391. 

L363 do you use air or nitrogen?? Please be precise!

R21. Thank you for pointing out this discrepancy. In Section 3.7 (Line 381) air was replaced by nitrogen. 

L368 Liver has a density around 1.079 g/cm³ why is your 45 cm³ piece so heavy?

R22. Thank you for pointing out this typo. The weight of the 45 cm3 liver-agar sample was 46 grams. We adjusted the text accordingly in Section 3.7 (Line 398). 

L415 The discussion is far too long. Please focus on your main research question and do not discuss future projects. This is a paper not a Master’s thesis!

R23. Thank you for this suggestion. We removed the discussion on the future project regarding the control of the needle. 

L559 55 ref are more than enough…

R24. In our opinion, these references are essential to underpin the claims in our manuscript, ensuring the reliability of the information presented. 

Fig 1 give the reader an impression of size do not rely on colours; Fig 2 also misses a scale; Fig 5 b misses a scale; Fig 9 a misses a scale; Fig 10 b misses a scale.

R25. Thank you for these suggestions. To give the reader an impression of the size in the images, we have added scale bars in Figs 1, 2b, 5b, 9abc, 10c. 

Reviewer #2: The paper proposes a novel self-propelled needle designed for MRI-guided percutaneous laser ablation of the prostate. The self-propelling mechanism is inspired by the wasp’s ovipositor and consists of six needle segments that are driven by pneumatic actuators independently. The newly developed needle was tested in a gel phantom and an ex vivo tissue. While the mechanism has been proposed previously, this work appears to be the first attempt to make such a mechanism work in the MRI environment.

The work offers a solid contribution to the research community. If successful, the proposed mechanism will allow placing a needle into the prostate under live MRI guidance more efficiently, as it addresses the issues associated with MRI guidance, such as the narrow workspace in the MRI gantry and the deviation of the needle due to tissue-needle interaction. The paper is well written with sufficient detail of the work.

However, the performance of the self-propelling mechanism, especially the slip ratio, is somewhat unconvincing, probably due to the lack of comprehensive understanding of the forces involved in the propelling. This reviewer still thinks that the experimental data are worth publishing, though the paper needs to address the following questions.

(1) (Major) Equation (1) only considers the cutting force and the friction between the needle shaft and the tissue, though it is not clear if they are the dominant forces when considering the propulsion of the needle. If the friction is dominant, the slip ratio in Figure 7 should have increased as the needle advances into the tissue because the friction should be proportional to the contact area (i.e., the length of the needle inside the tissue). However, the figures do not seem to show such an effect except S4-105.

R26. Thank you for your compliments. We extended Eq 1 with the surface stiffness force. Please refer to R2 above. Indeed, the friction is length-dependent; as the needle propels deeper into the tissue, the surface area of the needle segments in direct contact with the tissue increases, increasing the friction forces on the needle segment linearly with insertion depth while the cutting force remains roughly constant throughout the insertion.

At the moment, the slip ratio of the needle in the tissue (phantom) is relatively high, which makes it difficult to observe the effect of friction increase during needle insertion. However, in theory, this effect would be beneficial for the self-propelling motion of the needle. 

(2) Related to comment 1, did the authors measure the friction and cutting forces independently? They could have measured them by inserting and retracting the needle in the phantom while measuring the force along the needle. (Assuming that the thrust force required for insertion would equal the sum of the friction and the cutting force, while that for retraction would only equal the friction.)

R27. The friction forces and cutting forces were not measured in the experiments. However, we agree that this is an interesting experiment to perform to determine the cutting and friction forces acting on a single needle segment in the future. In Section 4.3 we added a line about this futur

---

## [Decision Letter · Decision Letter 1]

26 Mar 2024

PONE-D-23-38960R1Design and evaluation of a pneumatic actuation unit for a wasp-inspired self-propelled needlePLOS ONE

Dear Dr. Bloemberg,

Thank you for submitting your manuscript to PLOS ONE. After careful consideration, we feel that it has merit but does not fully meet PLOS ONE’s publication criteria as it currently stands. Therefore, we invite you to submit a revised version of the manuscript that addresses the points raised during the review process.

We look forward to receiving your revised manuscript.

Kind regards,

Baeckkyoung Sung, Ph.D.

Academic Editor

PLOS ONE

**Additional Editor Comments:**

The authors are recommended to respond to the comments of Reviewer 1 with an appropreate revision of the manuscript.

Reviewers' comments:

Reviewer's Responses to Questions

**Comments to the Author**

1. If the authors have adequately addressed your comments raised in a previous round of review and you feel that this manuscript is now acceptable for publication, you may indicate that here to bypass the “Comments to the Author” section, enter your conflict of interest statement in the “Confidential to Editor” section, and submit your "Accept" recommendation.

Reviewer #1: (No Response)

Reviewer #2: All comments have been addressed

2. Is the manuscript technically sound, and do the data support the conclusions?

Reviewer #1: Yes

Reviewer #2: Yes

3. Has the statistical analysis been performed appropriately and rigorously? 

Reviewer #1: N/A

Reviewer #2: N/A

4. Have the authors made all data underlying the findings in their manuscript fully available?

Reviewer #1: No

Reviewer #2: Yes

5. Is the manuscript presented in an intelligible fashion and written in standard English?

Reviewer #1: Yes

Reviewer #2: Yes

6. Review Comments to the Author

Reviewer #1: Thanks very much for this thorough revision of your paper. It reads much better, while still is very long. The focus is a bit clearer now.

You introduced a stiffness force in the equilibrium equation. However, this is dependent on the actual movement (F=k*x) and rising with stroke while the other forces are more or less constant (F_cut) or dependent on the insertion depth (F_fric). I understand that F_stiff is acting in the direction of movement, because otherwise it would be F_fric. Please explain.

Concerning you R17: I must insist that Young’s modulus is of lesser interest when characterizing the cutting or ripping properties of a tissue than the actual material strength. Young’s modulus is a material-specific elasticity, not a strength. It would show you how elastic a tissue behaves, but not how much force you need to cut it. Let me make a comparison: Aluminium (no matter which alloy) has a Young’s modulus of about 70000 N/mm² while steel has roughly 210000 N/mm². A simple construction steel (S235, 1.0038) has a maximum strength of 360 N/mm² while a high-end aluminium alloy like AlZnMgCu1,5 can bear up to 540 N/mm². Which material is stronger, which is more elastic?

About the state of the art. I still feel some kind of unhappy with the situation that there seem to be such limited number of groups working on that field. I found the group of Oliver Schwarz from Stuttgart. He is working on pneumatically-assisted tools based on the Sirex wasp ovipositor. However, these seem to be bigger than your device. In addition, I found https://doi.org/10.1177/09544119221137133, DOI 10.1088/1748-3190/acfb65 (it is SMA-actuated…)

The length of the paper is still exhausting. Especially the discussion (including limitations and recommendations is far too much. Please cut down to a max of 1.5 pages. This should not be a summary but a discussion.

Reviewer #2: All the previous comments from this reviewer have been addressed. The manuscript is recommended for publication.

7. PLOS authors have the option to publish the peer review history of their article (what does this mean?). If published, this will include your full peer review and any attached files.

Reviewer #1: No

Reviewer #2: No

---

## [Author Response · Author response to Decision Letter 1]

5 Apr 2024

Dear Baeckkyoung Sung, Ph.D., 

Thank you for your time and consideration of our manuscript. Based on the reviewers’ comments below and your editorial comments, we have revised the manuscript accordingly. Please find our detailed responses to the reviewers below. In the revised manuscript, the changes are indicated using track changes.

Review Comments to the Author

Reviewer #1: Thanks very much for this thorough revision of your paper. It reads much better, while still is very long. The focus is a bit clearer now.

R1. Thank you for your compliments on our revised manuscript. Regarding the length of the manuscript, please refer to R5. 

You introduced a stiffness force in the equilibrium equation. However, this is dependent on the actual movement (F=k*x) and rising with stroke while the other forces are more or less constant (F_cut) or dependent on the insertion depth (F_fric). I understand that F_stiff is acting in the direction of movement, because otherwise it would be F_fric. Please explain.

R2. In Equation 1, we introduce the surface stiffness force, the friction force, and the cutting force. Both the surface stiffness force and the cutting force arise at the needle tip, whereas the friction force occurs along the length of the needle. The surface stiffness is due to the elasticity of the tissue layer and occurs when puncturing the skin or a stiffer tissue layer than the current surrounding tissue, e.g., when puncturing the membrane around an organ. We extended the explanation of the surface stiffness force in Line 103. The cutting force arises at the needle tip and is, amongst others, caused by the needle still encountering stiffness as it cuts through the tissue. We clarified the explanation of the cutting force in Lines 110-111. 

Concerning you R17: I must insist that Young’s modulus is of lesser interest when characterizing the cutting or ripping properties of a tissue than the actual material strength. Young’s modulus is a material-specific elasticity, not a strength. It would show you how elastic a tissue behaves, but not how much force you need to cut it. Let me make a comparison: Aluminium (no matter which alloy) has a Young’s modulus of about 70000 N/mm² while steel has roughly 210000 N/mm². A simple construction steel (S235, 1.0038) has a maximum strength of 360 N/mm² while a high-end aluminium alloy like AlZnMgCu1,5 can bear up to 540 N/mm². Which material is stronger, which is more elastic?

R3. Thank you for your valuable suggestion. We agree with you that for evaluation of the propulsion of the needle inside tissue (phantoms), mechanical properties, such as the needle-tissue friction and the tissue (phantom) cutting and ripping properties, e.g., the ultimate strength of the tissue, are of higher interest than the Young’s modulus of the tissue (phantom). However, to our knowledge, the majority of the mechanical properties of human tissues, such as needle-tissue (phantom) friction and tissue (phantom) ultimate strength are not described in the scientific literature. Thus, we chose to report the known mechanical properties of the prostate tissue and the gelatin phantoms, namely the modulus of elasticity (Kelly et al., 2019). We clarified this in Lines 303-306.

In addition, in the discussion in Lines 478-483, we added the recommendation to investigate the mechanical properties of prostate tissue and the corresponding tissue-mimicking materials important for needle insertion and propulsion, such as the needle-tissue friction coefficient and tissue ultimate strength.

• Kelly NP, Flood HD, Hoey DA, Kiely PA, Giri SK, Coffey JC, et al. Direct mechanical characterization of prostate tissue—a systematic review. The Prostate. 2019;79(2):115-25. doi: 10.1002/pros.23718.

About the state of the art. I still feel some kind of unhappy with the situation that there seem to be such limited number of groups working on that field. I found the group of Oliver Schwarz from Stuttgart. He is working on pneumatically-assisted tools based on the Sirex wasp ovipositor. However, these seem to be bigger than your device. In addition, I found https://doi.org/10.1177/09544119221137133, DOI 10.1088/1748-3190/acfb65 (it is SMA-actuated…)

R4. Thank you for your suggestions. In the introduction, we added the references to the mosquito-inspired needles of Acharya et al. (2023) and Gidde et al. (2023) in Line 82 where we state that in the scientific literature, other sources of bioinspiration, such as mosquitoes, were used for needles that employ different working principles. 

• Acharya SR, Hutapea P. An experimental study on the mechanics and control of SMA-actuated bioinspired needle. Bioinspiration & Biomimetics. 2023;18(6):066008. doi: 10.1088/1748-3190/acfb65.

• Gidde STR, Islam S, Kim A, Hutapea P. Experimental study of mosquito-inspired needle to minimize insertion force and tissue deformation. Proceedings of the Institution of Mechanical Engineers, Part H: Journal of Engineering in Medicine. 2023;237(1):113-23. doi: 10.1177/09544119221137133.

Furthermore, in Lines 83-84, we have added that wasp-inspired propulsion has also been shown to be useful on a larger scale in a drilling device for medical applications (Nakajima and Schwarz, 2014).

• Nakajima K, Schwarz O. How to use the ovipositor drilling mechanism of Hymenoptera for developing a surgical instrument in biomimetic design. International Journal of Design & Nature and Ecodynamics. 2014;9(3):177-89. doi: 10.2495/DNE-V9-N3-177-189.

The length of the paper is still exhausting. Especially the discussion (including limitations and recommendations is far too much. Please cut down to a max of 1.5 pages. This should not be a summary but a discussion.

R5. Thank you for your suggestion. In response, we have streamlined our discussion by omitting the results of our experiments and the discussion slip ratio peaks in Section 4.1 Main findings. Additionally, we have removed the discussion on longer stroke lengths from Section 4.2 Limitations. Lastly, we condensed the content of Section 4.3 Recommendations and Future Research. 

Reviewer #2: All the previous comments from this reviewer have been addressed. The manuscript is recommended for publication.

R6. Thank you for your kind words.

Thank you for your useful suggestions. We look forward to your reaction.

With kind regards,

Jette Bloemberg and co-authors

---

## [Decision Letter · Decision Letter 2]

26 Apr 2024

PONE-D-23-38960R2Design and evaluation of a pneumatic actuation unit for a wasp-inspired self-propelled needlePLOS ONE

Dear Dr. Bloemberg,

Thank you for submitting your manuscript to PLOS ONE. After careful consideration, we feel that it has merit but does not fully meet PLOS ONE’s publication criteria as it currently stands. Therefore, we invite you to submit a revised version of the manuscript that addresses the points raised during the review process.

We look forward to receiving your revised manuscript.

Kind regards,

Baeckkyoung Sung, Ph.D.

Academic Editor

PLOS ONE

Additional Editor Comments:

The authors are recommended to additionally revise their manuscript based on the revierer's comments.

Reviewers' comments:

Reviewer's Responses to Questions

**Comments to the Author**

1. If the authors have adequately addressed your comments raised in a previous round of review and you feel that this manuscript is now acceptable for publication, you may indicate that here to bypass the “Comments to the Author” section, enter your conflict of interest statement in the “Confidential to Editor” section, and submit your "Accept" recommendation.

Reviewer #1: (No Response)

2. Is the manuscript technically sound, and do the data support the conclusions?

Reviewer #1: Yes

3. Has the statistical analysis been performed appropriately and rigorously? 

Reviewer #1: N/A

4. Have the authors made all data underlying the findings in their manuscript fully available?

Reviewer #1: Yes

5. Is the manuscript presented in an intelligible fashion and written in standard English?

Reviewer #1: Yes

6. Review Comments to the Author

Reviewer #1: Thanks again very much for the diligent revision of your paper. It has again improved, while it is still very long.

You cite Abolhassani 2007 with his paper on friction forces on needles. Why not have a drawing that illustrates the forces acting on the needle instead of relying on a simple equation. Abolhassani also gives some info on friction and stiffness in eqs. 2 and 3. As your device is strongly relying on friction this is worth a try.

Material parameters could be estimated from these papers:

* Ankersen 1999 - Puncture resistance and tensile strength of skin simulants

* Gallagher 2012 - Dynamic Tensile Properties of Human Skin (they also give some literature overview)

* Johnson 2021 - Characterizing the Material Properties of the Kidney and Liver in Unconfined Compression and Probing Protocols with Special Reference to Varying Strain Rate (Fig 5)

* Kelly 2018 - Direct mechanical characterization of prostate tissue—a systematic review (promising title, but no real info on strength, as they use elastography…)

* Kemper 2010 - Biomechanical Response of Human Liver in Tensile Loading

* Liang 2016 - Simulation and experiment of soft-tissue deformation in prostate brachytherapy (at least some info on shear modulus)

* Wang 2014 - Alterations in mechanical properties are associated with prostate cancer progression (material stregth only to a certain extend)

To solve this argument, you could either focus the paper more on the technical aspect of *building* the device and leave out the theory background or neglect the building (and being happy that it works) and focus on the theory *why* it works.

The length of the paper is still above average.

7. PLOS authors have the option to publish the peer review history of their article (what does this mean?). If published, this will include your full peer review and any attached files.

Reviewer #1: No

---

## [Author Response · Author response to Decision Letter 2]

8 May 2024

Dear Baeckkyoung Sung, Ph.D., 

Thank you for your time and consideration of our manuscript. Based on the reviewer’s comments below, we have revised the manuscript accordingly. Please find our detailed responses to the reviewer below. In the revised manuscript, the changes are indicated using track changes.

Review Comments to the Author

Reviewer #1: Thanks again very much for the diligent revision of your paper. It has again improved, while it is still very long.

R1. Thank you for your compliments on our revised manuscript. Regarding the length of the manuscript, please refer to R4.

You cite Abolhassani 2007 with his paper on friction forces on needles. Why not have a drawing that illustrates the forces acting on the needle instead of relying on a simple equation. Abolhassani also gives some info on friction and stiffness in eqs. 2 and 3. As your device is strongly relying on friction this is worth a try.

R2. In Figure 2A, we visualized the ovipositor-inspired needle using a schematic drawing. The drawing shows a two-dimensional illustration of ovipositor-inspired needle insertion into tissue. In the drawing, we illustrated the forces acting on the needle, i.e., the friction force along the advancing needle segment, the cutting force on the tip of the advancing needle segment, and the friction force along the retracting needle segments, which works in the opposite direction as the friction force of the advancing needle segments. 

Material parameters could be estimated from these papers:

* Ankersen 1999 - Puncture resistance and tensile strength of skin simulants

* Gallagher 2012 - Dynamic Tensile Properties of Human Skin (they also give some literature overview)

* Johnson 2021 - Characterizing the Material Properties of the Kidney and Liver in Unconfined Compression and Probing Protocols with Special Reference to Varying Strain Rate (Fig 5)

* Kelly 2018 - Direct mechanical characterization of prostate tissue—a systematic review (promising title, but no real info on strength, as they use elastography…)

* Kemper 2010 - Biomechanical Response of Human Liver in Tensile Loading

* Liang 2016 - Simulation and experiment of soft-tissue deformation in prostate brachytherapy (at least some info on shear modulus)

* Wang 2014 - Alterations in mechanical properties are associated with prostate cancer progression (material stregth only to a certain extend)

R3. Thank you for your suggestions on estimating the material parameters of tissue. We agree that for evaluating the propulsion of the needle inside tissue phantoms and ex vivo specimens, mechanical properties, such as the needle-tissue friction, shear modulus, and ultimate strength, are of interest. However, the material parameters of the skin, such as its puncture resistance (Ankersen, 1999) and its dynamic tensile properties (Gallagher, 2012), are not of interest to our needle in the current evaluation stage because the self-propelled motion of our ovipositor-inspired needle requires contact with surrounding tissue and is not used during the puncture phase. Therefore, it works in the post-puncture phase where the surface stiffness force, i.e., the pre-puncture force due to the elasticity of the skin layer, equals zero. Consequently, we evaluated our needle’s slip ratio after an initial insertion distance of 35 mm where the surface stiffness force does not play a role.

Nevertheless, we agree that the mechanical properties of tissue phantoms and ex vivo tissue specimens, such as the needle-tissue friction, shear modulus, and ultimate strength, are of interest. We clarified this in Lines 306 and 436-443 of our revised manuscript. Furthermore, when using ex vivo porcine tissue specimens, we added the important remark that material properties may vary from human tissue (Johnson, 2021). 

To solve this argument, you could either focus the paper more on the technical aspect of *building* the device and leave out the theory background or neglect the building (and being happy that it works) and focus on the theory *why* it works.

The length of the paper is still above average.

R4. According to the submission guidelines of PLOS ONE, there are no restrictions on word count for a manuscript. In our opinion, the background theory and the technical aspects of developing and producing the design are essential for the thoroughness and completeness of our manuscript. 

Thank you for your valuable suggestions. We look forward to your reaction.

With kind regards,

Jette Bloemberg and co-authors

---

## [Decision Letter · Decision Letter 3]

22 May 2024

PONE-D-23-38960R3Design and evaluation of a pneumatic actuation unit for a wasp-inspired self-propelled needlePLOS ONE

Dear Dr. Bloemberg,

Thank you for submitting your manuscript to PLOS ONE. After careful consideration, we feel that it has merit but does not fully meet PLOS ONE’s publication criteria as it currently stands. Therefore, we invite you to submit a revised version of the manuscript that addresses the points raised during the review process.

We look forward to receiving your revised manuscript.

Kind regards,

Baeckkyoung Sung, Ph.D.

Academic Editor

PLOS ONE

Journal Requirements:

**Additional Editor Comments:**

The authors are recommended to revise the manuscript in line with the reviewer's comments.

Reviewers' comments:

Reviewer's Responses to Questions

**Comments to the Author**

1. If the authors have adequately addressed your comments raised in a previous round of review and you feel that this manuscript is now acceptable for publication, you may indicate that here to bypass the “Comments to the Author” section, enter your conflict of interest statement in the “Confidential to Editor” section, and submit your "Accept" recommendation.

Reviewer #1: (No Response)

2. Is the manuscript technically sound, and do the data support the conclusions?

Reviewer #1: Yes

3. Has the statistical analysis been performed appropriately and rigorously? 

Reviewer #1: N/A

4. Have the authors made all data underlying the findings in their manuscript fully available?

Reviewer #1: Yes

5. Is the manuscript presented in an intelligible fashion and written in standard English?

Reviewer #1: Yes

6. Review Comments to the Author

Reviewer #1: Thanks again very much for the new revision of your paper. It is still very long. I agree that there is no page limit by PLOS1 but I am convinced, that keeping a paper as brief as possible is a quality. There is not much “background theory” in your paper apart from eq. 1. Also, you do not compare theoretically derived values (Forces, Stroke etc.) with those you measured (eg. Fig. 8). Thus, I suggest for the sake of brevity to skip the “background theory” – or provide a real model describing the expected behaviour of your device… Don’t get me wrong: you built a great device. Wonderful engineering and outstanding experimental work. But you fail to explain, why it works the way it does quantitatively. Anyway. Keep it as is. Reviewers shouldn’t argue with authors and vice versa.

In eq. 1 you list all forces acting on the needle while in Fig. 2a F_{stiff} is missing. In your rebuttal you explain that this force is not necessary as the needle acts only deep in the tissue and the stiffness would act only while puncturing the tissue. I found this explanation on your manuscript in lines 103 ff but not in connection with eq. 1. Consequently, the reader would miss F_{stiff}. In addition. As the tissue is elastic AND friction acts, I would assume that after most of the accumulated F_{stiff} is released after puncturing the surface, there would still be a residual part of F_{stiff} acting. But I might be wrong. Still, if F_{stiff} appears in eq. 1 it must be defined or explained in Fig 2a.

7. PLOS authors have the option to publish the peer review history of their article (what does this mean?). If published, this will include your full peer review and any attached files.

Reviewer #1: No

---

## [Author Response · Author response to Decision Letter 3]

13 Jun 2024

Dear Baeckkyoung Sung, Ph.D., 

Thank you for your time and consideration of our manuscript. Based on the reviewer’s comments below, we have revised the manuscript accordingly. Please find our detailed responses to the reviewer below. In the revised manuscript, the changes are indicated using track changes.

Review Comments to the Author

Reviewer #1: Thanks again very much for the new revision of your paper. It is still very long. I agree that there is no page limit by PLOS1 but I am convinced, that keeping a paper as brief as possible is a quality. There is not much “background theory” in your paper apart from eq. 1. Also, you do not compare theoretically derived values (Forces, Stroke etc.) with those you measured (eg. Fig. 8). Thus, I suggest for the sake of brevity to skip the “background theory” – or provide a real model describing the expected behaviour of your device… Don’t get me wrong: you built a great device. Wonderful engineering and outstanding experimental work. But you fail to explain, why it works the way it does quantitatively. Anyway. Keep it as is. Reviewers shouldn’t argue with authors and vice versa.

R1. Thank you for your compliments on our revised manuscript. 

In eq. 1 you list all forces acting on the needle while in Fig. 2a F_{stiff} is missing. In your rebuttal you explain that this force is not necessary as the needle acts only deep in the tissue and the stiffness would act only while puncturing the tissue. I found this explanation on your manuscript in lines 103 ff but not in connection with eq. 1. Consequently, the reader would miss F_{stiff}. In addition. As the tissue is elastic AND friction acts, I would assume that after most of the accumulated F_{stiff} is released after puncturing the surface, there would still be a residual part of F_{stiff} acting. But I might be wrong. Still, if F_{stiff} appears in eq. 1 it must be defined or explained in Fig 2a.

R2. Thank you for your suggestion. In the figure caption of Fig 2A, we added that this is an illustration of the needle in the post-puncture phase after the needle has penetrated the outer tissue layer, therefore, F_stiff is not present. Furthermore, in Lines 107-109 and 131-132, we explained that F_stiff is only present when puncturing tissue layers. After the needle has punctured the tissue layer, F_stiff returns to zero whilst self-propelling through homogeneous tissue. 

Thank you for your valuable suggestions. We look forward to your reaction.

With kind regards,

Jette Bloemberg and co-authors

---

## [Decision Letter · Decision Letter 4]

18 Jun 2024

Design and evaluation of a pneumatic actuation unit for a wasp-inspired self-propelled needle

PONE-D-23-38960R4

Dear Dr. Bloemberg,

We’re pleased to inform you that your manuscript has been judged scientifically suitable for publication and will be formally accepted for publication once it meets all outstanding technical requirements.

Kind regards,

Baeckkyoung Sung, Ph.D.

Academic Editor

PLOS ONE

Additional Editor Comments (optional):

All comments and requirements raised by the reviewers have been fully reflected in the revised manuscript.

Reviewers' comments:

Reviewer's Responses to Questions

**Comments to the Author**

1. If the authors have adequately addressed your comments raised in a previous round of review and you feel that this manuscript is now acceptable for publication, you may indicate that here to bypass the “Comments to the Author” section, enter your conflict of interest statement in the “Confidential to Editor” section, and submit your "Accept" recommendation.

Reviewer #1: All comments have been addressed

2. Is the manuscript technically sound, and do the data support the conclusions?

Reviewer #1: Yes

3. Has the statistical analysis been performed appropriately and rigorously? 

Reviewer #1: Yes

4. Have the authors made all data underlying the findings in their manuscript fully available?

Reviewer #1: Yes

5. Is the manuscript presented in an intelligible fashion and written in standard English?

Reviewer #1: Yes

6. Review Comments to the Author

Reviewer #1: Thanks very much for your revision. The form asks me to write at least 100 characters, so I continue until this limit is reac...

7. PLOS authors have the option to publish the peer review history of their article (what does this mean?). If published, this will include your full peer review and any attached files.

Reviewer #1: No

---

## [Editor Report · Acceptance letter]

23 Jun 2024

PONE-D-23-38960R4 

PLOS ONE

Dear Dr. Bloemberg, 

I'm pleased to inform you that your manuscript has been deemed suitable for publication in PLOS ONE. Congratulations! Your manuscript is now being handed over to our production team.

Kind regards, 

on behalf of

Dr. Baeckkyoung Sung 

Academic Editor

PLOS ONE